# The CoT COLLECTION: Improving Zero-shot and Few-shot Learning of Language Models via Chain-of-Thought Fine-Tuning

**Seungone Kim**[1,2*†]     **Se June Joo**[1*†]     **Doyoung Kim**[1]     **Joel Jang**[1,3]
**Seonghyeon Ye**[1]     **Jamin Shin**[2†]     **Minjoon Seo**[1†]

KAIST AI[1]     NAVER AI Lab[2]     University of Washington[3]
{seungone, sejune, minjoon}@kaist.ac.kr     jayshin.nlp@gmail.com

## Abstract

Language models (LMs) with less than 100B parameters are known to perform poorly on chain-of-thought (CoT) reasoning in contrast to large LMs when solving unseen tasks. In this work, we aim to equip smaller LMs with the step-by-step reasoning capability by instruction tuning with CoT rationales. In order to achieve this goal, we first introduce a new instruction-tuning dataset called the CoT COLLECTION, which augments the existing Flan Collection (including only 9 CoT tasks) with additional 1.84 million rationales across 1,060 tasks. We show that CoT fine-tuning Flan-T5 (3B & 11B) with CoT COLLECTION enables smaller LMs to have better CoT capabilities on unseen tasks. On the BIG-Bench-Hard (BBH) benchmark, we report an average improvement of +4.34% (Flan-T5 3B) and +2.60% (Flan-T5 11B), in terms of zero-shot task accuracy. Furthermore, we show that instruction tuning with CoT COLLECTION allows LMs to possess stronger few-shot learning capabilities on 4 domain-specific tasks, resulting in an improvement of +2.24% (Flan-T5 3B) and +2.37% (Flan-T5 11B), even outperforming ChatGPT utilizing demonstrations until the max length by a +13.98% margin. Our code, the CoT COLLECTION data, and model checkpoints are publicly available [1].

## 1 Introduction

Language models (LMs) pre-trained on massive text corpora can adapt to downstream tasks in both zero-shot and few-shot learning settings by incorporating task instructions and demonstrations (Brown et al., 2020; Wei et al., 2021; Sanh et al., 2021; Mishra et al., 2022; Wang et al., 2022b; Iyer et al., 2022; Liu et al., 2022b; Chung et al., 2022; Longpre et al., 2023; Ye et al., 2023). One approach that

has been particularly effective in enabling LMs to excel at a multitude of tasks is Chain-of-Thought (CoT) prompting, making LMs generate a rationale to derive its final prediction in a sequential manner (Wei et al., 2022b; Kojima et al., 2022; Zhou et al., 2022; Zhang et al., 2022; Yao et al., 2023).

While CoT prompting works effectively for large LMs with more than 100 billion parameters, it does not necessarily confer the same benefits to smaller LMs (Tay et al., 2022; Suzgun et al., 2022; Wei et al., 2022a; Chung et al., 2022). The requirement of a large number of parameters consequently results in significant computational cost and accessibility issues (Kaplan et al., 2020; Min et al., 2022; Liu et al., 2022b; Mhlanga, 2023; Li et al., 2023).

Recent work has focused on empowering relatively smaller LMs to effectively solve novel tasks as well, primarily through fine-tuning with rationales (denoted as CoT fine-tuning) and applying CoT prompting on a single target task (Shridhar et al., 2022; Ho et al., 2022; Fu et al., 2023). However, solving a *single* task does not adequately address the issue of generalization to a broad range of unseen tasks. While Chung et al. (2022) leverage 9 publicly available CoT tasks during instruction tuning to solve multiple unseen tasks, the imbalanced ratio compared to 1,827 tasks used for direct fine-tuning results in poor CoT results across smaller LMs (Longpre et al., 2023). In general, the community still lacks a comprehensive strategy to fully leverage CoT prompting to solve *multiple* unseen novel tasks in the context of smaller LMs.

To bridge this gap, we present the CoT COLLECTION, an instruction tuning dataset that augments 1.84 million rationales from the FLAN Collection (Longpre et al., 2023) across 1,060 tasks. We fine-tune Flan-T5 (3B & 11B) using CoT COLLECTION and denote the resulting model as CoT-T5. We perform extensive comparisons of CoT-T5 and Flan-T5 under two main scenarios: (1) zero-shot learning and (2) few-shot learning.

---

*denotes equal contribution. Work was done while Seungone was interning at NAVER AI Lab.

†Corresponding authors

[1]https://github.com/kaistAI/
CoT-Collection

In the *zero-shot* learning setting, CoT-T5 (3B & 11B) outperforms Flan-T5 (3B & 11B) by +4.34% and +2.60% on average accuracy across 27 datasets from the Big Bench Hard (BBH) benchmark (Suzgun et al., 2022) when evaluated with CoT prompting. During ablation experiments, we show that CoT fine-tuning T0 (3B) (Sanh et al., 2021) on a subset of the CoT Collection, specifically 163 training tasks used in T0, shows a performance increase of +8.65% on average accuracy across 11 datasets from the P3 Evaluation benchmark. Moreover, we translate 80K instances of CoT COLLECTION into 5 different languages (French, Japanese, Korean, Russian, Chinese) and observe that CoT fine-tuning mT0 (3B) (Muennighoff et al., 2022) on each language results in 2x ∼ 10x performance improvement on average accuracy across all 5 languages from the MGSM benchmark (Shi et al., 2022).

In the *few-shot learning* setting, where LMs must adapt to new tasks with a minimal number of instances, CoT-T5 (3B & 11B) exhibits a +2.24% and +2.37% improvement on average compared to using Flan-T5 (3B & 11B) as the base model on 4 different domain-specific tasks[2]. Moreover, it demonstrates +13.98% and +8.11% improvement over ChatGPT (OpenAI, 2022) and Claude (Anthropic, 2023) that leverages ICL with demonstrations up to the maximum input length.

Our contributions are summarized as follows:

- We introduce CoT COLLECTION, a new instruction dataset that includes 1.84 million rationales across 1,060 tasks that could be used for applying CoT fine-tuning to LMs.

- With CoT COLLECTION, we fine-tune Flan-T5, denoted as CoT-T5, which shows a nontrivial boost in zero-shot and few-shot learning capabilities with CoT Prompting.

- For ablations, we show that CoT fine-tuning could improve the CoT capabilities of LMs in low-compute settings by using a subset of CoT COLLECTION and training on (1) smaller number of tasks (T0 setting; 163 tasks) and (2) smaller amount of instances in 5 different languages (French, Japanese, Korean, Russian, Chinese; 80K instances).

---

[2]We assess the efficacy of our approach on 4 domain-specific datasets, two each from legal and medical fields, namely including LEDGAR (Tuggener et al., 2020), Case Hold (Zheng et al., 2021), MedNLI (Romanov and Shivade, 2018), and PubMedQA (Jin et al., 2019). Each dataset is represented by 64 randomly chosen instances.

## 2 Related Works

### 2.1 Chain-of-Thought (CoT) Prompting

Wei et al. (2022b) propose Chain of Thought (CoT) Prompting, a technique that triggers the model to generate a rationale before the answer. By generating a rationale, large LMs show improved reasoning abilities when solving challenging tasks. Kojima et al. (2022) show that by appending the phrase 'Let's think step by step', large LMs could perform CoT prompting in a zero-shot setting. Different work propose variants of CoT prompting such as automatically composing CoT demonstrations (Zhang et al., 2022) and performing a finegrained search through multiple rationale candidates with a tree search algorithm (Yao et al., 2023). While large LMs could solve novel tasks with CoT Prompting, Chung et al. (2022) and Longpre et al. (2023) show that this effectiveness does not necessarily hold for smaller LMs. In this work, we aim to equip smaller LMs with the same capabilities by instruction tuning on large amount of rationales.

### 2.2 Improving Zero-shot Generalization

Previous work show that instruction tuning enables generalization to multiple unseen tasks (Wei et al., 2021; Sanh et al., 2021; Aribandi et al., 2021; Ouyang et al., 2022; Wang et al., 2022b; Xu et al., 2022). Different work propose to improve instruction tuning by enabling cross-lingual generalization (Muennighoff et al., 2022), improving label generalization capability (Ye et al., 2022), and training modular, expert LMs (Jang et al., 2023). Meanwhile, a line of work shows that CoT fine-tuning could improve the reasoning abilities of LMs on a single-seen task (Zelikman et al., 2022; Shridhar et al., 2022; Ho et al., 2022; Fu et al., 2023). As a follow-up study, we CoT fine-tune on 1,060 instruction tasks and observe a significant improvement in terms of zero-shot generalization on multiple tasks.

### 2.3 Improving Few-Shot Learning

For adapting LMs to new tasks with a few instances, recent work propose advanced parameter efficient fine-tuning (PEFT) methods, where a small number of trainable parameters are added (Hu et al., 2021; Lester et al., 2021; Liu et al., 2021, 2022b; Asai et al., 2022; Liu et al., 2022c). In this work, we show that a simple recipe of (1) applying LoRA (Hu et al., 2021) to a LM capable of performing CoT reasoning and (2) CoT fine-tuning on a target task results in strong few-shot performance.

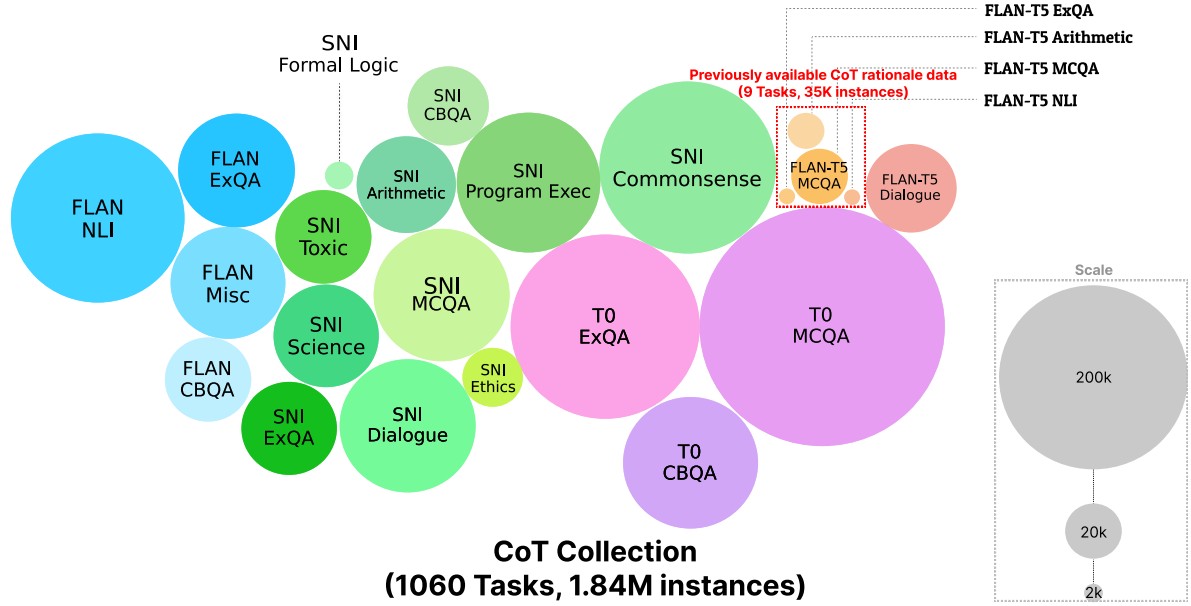

Figure 1: An illustration of the overall task group and dataset source of where we obtained the instances to augment the rationales in CoT COLLECTION. Compared to the 9 datasets that provide publicly available rationales (included within 'Flan-T5 ExQA', 'Flan-T5 Arithmetic', 'Flan-T5 MCQA', 'Flan-T5 NLI' from the red box), we generate ~51.29 times more rationales (1.84 million rationales) and ~117.78 times more task variants (1,060 tasks).

## 3 The CoT COLLECTION

Despite its effectiveness for CoT fine-tuning, rationale data still remains scarce. To the best of our knowledge, recent work mostly rely on 9 publicly available NLP datasets[3] for fine-tuning with rationales (Zelikman et al., 2022; Shridhar et al., 2022; Chung et al., 2022; Ho et al., 2022; Longpre et al., 2023; Fu et al., 2023). This is due to the difficulty in gathering human-authored rationales (Kim et al., 2023). To this end, we create CoT COLLECTION, an instruction-tuning dataset that includes 1.84 million rationales augmented across 1,060 tasks[4]. In this section, we explain the datasets we select to augment into rationales and how we perform the overall augmentation process.

**Broad Overview**  Given an input $X = [I, z]$ composed of an instruction $I$, and an instance $z$ along with the answer $y$, we obtain a rationale $r$ by applying in-context learning (ICL) with a large LM. Note that this differs from previous works which focused on generating new instances $z$ using large LMs (West et al., 2022; Liu et al., 2022a; Kim et al., 2022; Honovich et al., 2022; Wang et al., 2022a; Taori et al., 2023; Chiang et al., 2023) while we extend it to generating new rationales $r$.

**Source Dataset Selection**  As a source dataset to extract rationales, we choose the Flan Collection (Longpre et al., 2023), consisting of 1,836 diverse NLP tasks from P3 (Sanh et al., 2021), Super-NaturalInstructions (Wang et al., 2022b), Flan (Wei et al., 2021), and some additional dialogue & code datasets. We choose 1,060 tasks, narrowing our focus following the criteria as follows:

- Generation tasks with long outputs are excluded since the total token length of appending $r$ and $y$ exceeds the maximum output token length (512 tokens) during training.

- Datasets that are not publicly available such as DeepMind Coding Contents and Dr Repair (Yasunaga and Liang, 2020) are excluded.

- Datasets where the input and output do not correspond to each other in the huggingface datasets (Lhoest et al., 2021) are excluded.

- When a dataset appears in common across different sources, we prioritize using the task from P3 first, followed by SNI, and Flan.

- During preliminary experiments, we find that for tasks such as sentiment analysis, sentence

---

[3]The 9 available datasets are QASC (Khot et al., 2020), AQuA (Amini et al., 2019), GSM8K (Cobbe et al., 2021), QED (Lamm et al., 2021), StrategyQA (Geva et al., 2021), SenseMaking (Wang et al., 2019), CREAK (Onoe et al., 2021), e-SNLI (Camburu et al., 2018), ECQA (Aggarwal et al., 2021).

[4]Following Sanh et al. (2021), we use the notion of 'task' referring to each prompt applied to a dataset.

completion, coreference resolution, and word disambiguation, rationales generated by large LMs are very short and uninformative. We exclude these tasks to prevent negative transfer during multitask learning (Aribandi et al., 2021; Jang et al., 2023).

**Creating Demonstrations for ICL** We first create prompts to apply in-context learning (ICL) with large LMs for augmenting the instances in the selected tasks with rationales. Preparing demonstrations $\mathcal{D}^t$ for each task $t$ is the most straightforward, but it becomes infeasible to prepare demonstrations for each task as the number of tasks gets larger. Instead, we assign each task $t$ to $T_k$, a family of tasks that shares a similar task format such as multiple choice QA, closed book QA, and dialogue generation. Each family of tasks share $\mathcal{D}^{T_k}$, which consists of $6 \sim 8$ demonstrations. These $6 \sim 8$ demonstrations for each task group $T_k$ is manually created by 3 of the authors in this paper. Specifically, given 136 instances sampled from Flan Collection, two annotators are assigned to write a rationale, and the other third annotator conducts an A/B testing between the two options. We manually create $\mathcal{D}^{T_k}$ across $k = 26$ task groups. We include the prompts for all of the different task groups in Appendix D.

**Rationale Augmentation** We use the OpenAI Codex[5] to augment rationales. Formally, given $(X_i^t, y_i^t)$, the $i^{th}$ instance of a task $t$, the goal is to generate corresponding rationale $r_i^t$. Note that during preliminary experiments, we found that ordering the label in front of the rationale within the demonstration $\mathcal{D}^{T_k}$ was crucial to generate good quality rationales. We conjecture this is because ordering the label in front of the rationale loosens the need for the large LM to solve the underlying task and only focus on generating a rationale. However, we also found that in some tasks such as arithmetic reasoning, large LMs fail to generate good-quality rationales. To mitigate this issue, we apply filtering to the augmented rationales. We provide the criteria used for the filtering phase and

---

[5]The use of Codex was largely due to limited academic budget (OpenAI supported Codex with no cost for researchers up to June 2023). Moreover, other LLM services such as Bard (Google, 2023) and Claude (Anthropic, 2023) were not supported during the period of COT COLLECTION augmentation. To address the concern of reproducibility, analysis on quality of rationales from Codex, Bard and Claude is included in Appendix A,

Figure 2: MCQA Prompt used to augment rationales from P3 dataset. Through ICL, the large LM generates a rationale that is conditioned on the ground-truth label.

the filtered cases at Appendix B. Also, we include analysis of the diversity and quality of COT COLLECTION compared to the existing 9 CoT tasks and human-authored rationales in Appendix A.

## 4 Experiments

For our main experiments, we use Flan-T5 (Chung et al., 2022) as our base model, and obtain CoT-T5 by CoT fine-tuning on the COT COLLECTION. Formally, given $X_i^t$, the goal of CoT fine-tuning is to sequentially generate the rationale $r_i^t$ and answer $y_i^t$. To indicate that $r_i^t$ should be generated before $y_i^t$, the trigger phrase 'Let's think step by step' is added during both training and evaluation. We mostly follow the details for training and evaluation from Chung et al. (2022), and provide additional details in Appendix C. In this section, we show how training on COT COLLECTION enhances zero-shot generalization capabilities (Section 4.2) and few-shot adaptation capabilities (Section 4.3).

### 4.1 Evaluation

We evaluate under two different evaluation methods: **Direct Evaluation** and **CoT Evaluation**. For Direct Evaluation on classification tasks, we follow previous works using verbalizers, choosing the option with the highest probability through comparison of logit values (Schick and Schütze, 2021;

Sanh et al., 2021; Ye et al., 2022; Jang et al., 2023), and measure the accuracy. For generation tasks, we directly compare the LM's prediction with the answer and measure the EM score.

When evaluating with CoT Evaluation, smaller LMs including Flan-T5 often do not generate any rationales even with the trigger phrase 'Let's think step by step'. Therefore, we adopt a hard constraint of requiring the LM to generate $r_i^t$ with at least a minimum length of 8 tokens. In classification tasks, we divide into two steps where the LM first generates $r_i^t$, and then verbalizers are applied with a indicator phrase '[ANSWER]' inserted between $r_i^t$ and the possible options. For generation tasks, we extract the output coming after the indicator phrase. Accuracy metric is used for classification tasks while EM metric is used for generation tasks.

## 4.2 Zero-shot Generalization

In this subsection, we show how training with CoT COLLECTION could effectively improve the LM's ability to solve unseen tasks. We have three difference experimental set-ups, testing different aspects: **Setup #1**: training on the entire 1060 tasks in CoT COLLECTION and evaluating the reasoning capabilities of LMs with the Bigbench Hard (BBH) benchmark (Suzgun et al., 2022), **Setup #2**: training only on 163 tasks that T0 (Sanh et al., 2021) used for training (a subset of the CoT COLLECTION), and evaluating the linguistic capabilities of LMs with the P3 evaluation benchmark (Sanh et al., 2021), and **Setup #3**: training with a translated, subset version of CoT COLLECTION for each five different languages and evaluating how LMs could perform CoT reasoning in multilingual settings using the MGSM benchmark (Shi et al., 2022).

**Setup #1: CoT Fine-tuning with 1060 CoT Tasks** We first perform experiments with our main model, CoT-T5, by training Flan-T5 on the entire CoT COLLECTION and evaluate on the BBH benchmark (Suzgun et al., 2022). In addition to evaluating Flan-T5, we compare the performances of different baselines such as (1) T5-LM (Raffel et al., 2020): the original base model of Flan-T5, (2) T0 (Sanh et al., 2021): an instruction-tuned LM trained with P3 instruction dataset, (3) Tk-Instruct (Wang et al., 2022b): an instruction-tuned LM trained with SNI instruction dataset, and (4) GPT-3 (Brown et al., 2020): a pre-trained LLM with 175B parameters. For ablation purposes, we also train T5-LM with CoT COLLECTION (de-

| Method | CoT | Direct | Total Avg |
|---|---|---|---|
| T5-LM-3B | 26.68 | 26.96 | 26.82 |
| T0-3B | 26.64 | 27.45 | 27.05 |
| TK-INSTRUCT-3B | 29.86 | 29.90 | 29.88 |
| TK-INSTRUCT-11B | 33.60 | 30.71 | 32.16 |
| T0-11B | 31.83 | 33.57 | 32.70 |
| FLAN-T5-3B | 34.06 | 37.14 | 35.60 |
| GPT-3 (175B) | 38.30 | 33.60 | 38.30 |
| FLAN-T5-11B | 38.57 | _40.99_ | _39.78_ |
| T5-3B + CoT FT | 37.95 | 35.52 | 36.74 |
| CoT-T5-3B | 38.40 | 36.18 | 37.29 |
| T5-11B + CoT FT | _40.02_ | 38.76 | 39.54 |
| CoT-T5-11B | **42.20** | **42.56** | **42.38** |

Table 1: Evaluation performance on all the 27 unseen datasets from BBH benchmark, including generation tasks. All evaluations are held in a zero-shot setting. The best comparable performances are **bolded** and second best underlined.

noted as 'T5 + CoT FT'). Note that FLAN Collection includes 15 million instances, hence ∼8 times larger compared to our CoT COLLECTION.

The results on BBH benchmark are shown across Table 1 and Table 2. In Table 1, CoT-T5 (3B & 11B) achieves a +4.34% and +2.60% improvement over Flan-T5 (3B & 11B) with CoT Evaluation. Surprisingly, while CoT-T5-3B CoT performance improves +4.34% with the cost of 0.96% degradation in Direct Evalution, CoT-T5-11B's Direct Evaluation performance even improves, resulting in a +2.57% total average improvement. Since CoT COLLECTION only includes instances augmented with rationales, these results show that CoT fine-tuning could improve the LM's capabilities regardless of the evaluation method. Also, T5-3B + CoT FT and T5-11B + CoT FT outperforms FLAN-T5-3B and FLAN-T5-11B by a +1.45% and +3.89% margin, respectively, when evaluated with CoT evaluation. Moreover, T5-3B + CoT FINE-TUNING outperforms ∼4 times larger models such as T0-11B and Tk-Instruct-11B in both Direct and CoT Evaluation. The overall results indicate that (1) CoT fine-tuning on a diverse number of tasks enables smaller LMs to outperform larger LMs and (2) training with FLAN Collection and CoT Collection provides complementary improvements to LMs under different evaluation methods; CoT-T5 obtains good results across both evaluation methods by training on both datasets.

In Table 2, CoT-T5-11B obtains same or better results on 15 out of 23 tasks when evaluated with Direct evaluation, and 17 out of 23 tasks when evaluated with CoT Evaluation compared to Flan-

| Task | CoT-T5-11B | | FLAN-T5-11B | | VICUNA-13B | CHATGPT | CODEX | GPT-4 |
|------|------|------|------|------|------|------|------|------|
| | CoT | Direct | CoT | Direct | Direct | Direct | Direct | Direct |
| BOOLEAN EXPRESSIONS | **65.6** | 59.2 | 51.6 | 56.8 | 40.8 | 82.8 | 88.4 | 77.6 |
| CAUSAL JUDGMENT | 60.4 | 60.2 | 58.3 | **61.0** | 42.2 | 57.2 | 63.6 | 59.9 |
| DATE UNDERSTANDING | 52.0 | 51.0 | 46.8 | **54.8** | 10.0 | 42.8 | 63.6 | 74.8 |
| DISAMBIGUATION QA | 63.4 | **68.2** | 63.2 | 67.2 | 18.4 | 57.2 | 67.2 | 69.2 |
| FORMAL FALLACIES | 51.2 | **55.2** | 54.4 | **55.2** | 47.2 | 53.6 | 52.4 | 64.4 |
| GEOMETRIC SHAPES | **22.0** | 10.4 | 12.4 | 21.2 | 3.6 | 25.6 | 32.0 | 40.8 |
| HYPERBATON | 65.2 | 64.2 | 55.2 | **70.8** | 44.0 | 69.2 | 60.4 | 62.8 |
| LOGICAL DEDUCTION (5) | 48.2 | **54.4** | 51.2 | 53.6 | 4.8 | 38.8 | 32.4 | 66.8 |
| LOGICAL DEDUCTION (7) | 52.4 | **60.6** | 57.6 | 60.0 | 1.2 | 39.6 | 26.0 | 66.0 |
| LOGICAL DEDUCTION (3) | 55.4 | **75.0** | 66.4 | 74.4 | 16.8 | 60.4 | 52.8 | 94.0 |
| MOVIE RECOMMENDATION | 44.6 | **52.8** | 32.4 | 36.4 | 43.4 | 55.4 | 84.8 | 79.5 |
| NAVIGATE | 59.0 | 60.0 | 60.8 | **61.6** | 46.4 | 55.6 | 50.4 | 68.8 |
| PENGUINS IN A TABLE | 39.1 | **41.8** | **41.8** | **41.8** | 15.1 | 45.9 | 66.4 | 76.7 |
| REASONING COLORED OBJ. | 32.6 | **33.2** | 22.8 | 23.2 | 12.0 | 47.6 | 67.6 | 84.8 |
| RUIN NAMES | **42.8** | 41.6 | 31.6 | 34.4 | 15.7 | 56.0 | 75.2 | 89.1 |
| SALIENT TRANS ERR. | 43.8 | **49.2** | 35.6 | **49.2** | 2.0 | 40.8 | 62.0 | 62.4 |
| SNARKS | 67.7 | 66.2 | 59.5 | **70.2** | 28.1 | 59.0 | 61.2 | 87.6 |
| SPORTS UNDERSTANDING | 64.8 | **66.4** | 56.0 | 60.0 | 48.4 | 79.6 | 72.8 | 84.4 |
| TEMPORAL SEQUENCES | 27.4 | **28.8** | 24.4 | **28.8** | 16.0 | 35.6 | 77.6 | 98.0 |
| TRACKING SHUFF OBJ. (5) | **20.0** | 13.2 | 19.6 | 15.2 | 9.2 | 18.4 | 20.4 | 25.2 |
| TRACKING SHUFF OBJ. (7) | **18.4** | 9.6 | 13.2 | 12.0 | 5.6 | 15.2 | 14.4 | 25.2 |
| TRACKING SHUFF OBJ. (3) | **41.8** | 31.2 | 28.8 | 24.4 | 23.2 | 31.6 | 37.6 | 42.4 |
| WEB OF LIES | **57.0** | 51.6 | 52.8 | 50.0 | 41.2 | 56.0 | 51.6 | 49.6 |
| AVERAGE | 47.60 | 48.00 | 43.32 | 47.05 | 23.30 | 48.90 | 52.80 | 67.40 |

Table 2: Evaluation performance on 23 unseen classification datasets from BBH benchmark. Scores of Vicuna, ChatGPT, Codex (teacher model of CoT-T5), GPT-4 are obtained from Chung et al. (2022) and Mukherjee et al. (2023). Evaluations are held in a zero-shot setting. The best comparable performances are **bolded** and second best underlined among the open-sourced LMs.

| Method | Natural Language Inference | | | | | Sentence Completion | | | Coreference Resolut. | | WSD | Total Avg |
|--------|------|------|------|------|------|------|------|------|------|------|------|------|
| | RTE | CB | AN. R1 | AN. R2 | AN. R3 | COPA | Hellasw. | StoryC. | Winogr. | WSC | WiC | |
| T5-3B (Raffel et al., 2020) | 53.03 | 34.34 | 32.89 | 33.76 | 33.82 | 54.88 | 27.00 | 48.16 | 50.64 | 54.09 | 50.30 | 42.99 |
| T0-3B (SANH ET AL., 2021) | 60.61 | 48.81 | 35.10 | 33.27 | 33.52 | 75.13 | 27.18 | 84.91 | 50.91 | 65.00 | 51.27 | 51.43 |
| RoE-3B (JANG ET AL., 2023) | 64.01 | 43.57 | 35.49 | 34.64 | 31.22 | 79.25 | 34.60 | 86.33 | 61.60 | 62.21 | 52.97 | 53.48 |
| KiC-770M (PAN ET AL., 2022) | 74.00 | 67.90 | 36.30 | 35.00 | 37.60 | 85.30 | 29.60 | 94.40 | 55.30 | 65.40 | 52.40 | 57.56 |
| FLIPPED-3B (YE ET AL., 2022) | 71.05 | 57.74 | 39.99 | 37.05 | 37.73 | 89.88 | 41.64 | **95.88** | 58.56 | 58.37 | 50.42 | 58.03 |
| GPT-3 (175B) (BROWN ET AL., 2020) | 63.50 | 46.40 | 34.60 | 35.40 | 34.50 | **91.00** | **78.90** | 83.20 | **70.20** | 65.40 | 45.92 | 59.00 |
| T0-11B (SANH ET AL., 2021) | **80.83** | 70.12 | **43.56** | **38.68** | 41.26 | 90.02 | 33.58 | 92.40 | 59.94 | 61.45 | 56.58 | **60.76** |
| T5-3B + CoT FT - EVAL W/ DIRECT | 69.96 | 58.69 | 37.58 | 36.00 | 37.44 | 84.59 | 40.92 | 90.47 | 55.40 | 64.33 | 51.53 | 56.99 |
| T0-3B + CoT FT - EVAL W/ DIRECT | 80.79 | 65.00 | 39.49 | 35.13 | 38.58 | 88.27 | 41.04 | 92.13 | 56.40 | **65.96** | 53.60 | 59.67 |
| T5-3B + CoT FT - EVAL W/ CoT | 80.61 | 69.17 | 40.24 | 36.67 | 40.13 | 90.10 | 41.08 | 93.00 | 56.47 | 55.10 | **56.73** | 59.94 |
| T0-3B + CoT FT - EVAL W/ CoT | 80.25 | **72.62** | 41.71 | 37.22 | **41.89** | 90.88 | 39.50 | 94.47 | 57.47 | 50.58 | 54.27 | 60.08 |

Table 3: Evaluation performance on 11 different unseen P3 dataset (Sanh et al., 2021) categorized into 4 task categories. We report the direct performance of the baselines since they were not CoT fine-tuned on instruction data. The best comparable performances are **bolded** and second best underlined. We exclude Flan-T5 and CoT-T5 since they were trained on the unseen tasks (tasks from FLAN and SNI overlap with the P3 Eval datasets), breaking unseen task assumption.

T5-11B. Interestingly, Vicuna (Chiang et al., 2023), a LM trained on long-form dialogues between users and GPT models, perform much worse compared to both CoT-T5 and Flan-T5. We conjecture that training on instruction datasets from existing academic benchmarks consisting CoT Collection and Flan Collection is more effective in enabling LMs to solve reasoning tasks compared to chat LMs.

**Setup #2: CoT Fine-tuning with 163 CoT Tasks (T0 Setup)** To examine whether the effect of CoT fine-tuning is dependent on large number of tasks and instances, we use the P3 training subset from the CoT COLLECTION consisted of 644K instances from 163 tasks, and apply CoT fine-tuning to T0 (3B) (Sanh et al., 2021) and T5-LM (3B) (Raffel et al., 2020). Note that T0 is trained with 12M instances, hence ∼18.63 times larger. Then, we evaluate on the P3 evaluation benchmark which consists of 11 different NLP datasets. In addition to the baselines from the previous section (T5-LM, T0, and GPT-3), we also include LMs that are trained on the same T0 setup for comparison such as, (1) RoE (Jang et al., 2023): a modular expert LM that retrieves different expert models depending on the unseen task, (2) KiC (Pan et al., 2022): a retrieval-augmented model that

is instruction-tuned to retrieve knowledge from a KB memory, and (3) Flipped (Ye et al., 2022): an instruction-tuned model that is trained to generate the instruction in order to resolve the LM overfitting to the output label as baseline models.

The results are shown in Table 3. Surprisingly, T5-3B + CoT FT outperforms T0-3B by a +8.24% margin when evaluated with CoT Evaluation, while using ∼18.63 times less instances. This supports that CoT fine-tuning is *data efficient*, being effective even with less number of instances and tasks. Moreover, T0-3B + CoT FT improves T0-3B by +8.65% on average accuracy. When compared with T0-11B with ∼4 times more number of parameters, it achieves better performance at sentence completion, and word sense disambiguation (WSD) tasks, and obtains similar performances at natural language inference and coreference resolution tasks.

**Setup #3: Multilingual Adaptation with CoT Fine-tuning** In previous work, Shi et al. (2022) proposed MGSM, a multilingual reasoning benchmark composed of 10 different languages. In this subsection, we conduct a toy experiment to examine whether CoT fine-tuning could enable LMs to reason step-by-step in multilingual settings as well, using a subset of 5 languages (Korean, Russian, French, Chineses, Japanese) from MGSM.

In Table 4, current smaller LMs can be divided into three categories: (1) Flan-T5, a LM that is CoT fine-tuned with mostly English instruction data, (2) MT5 (Xue et al., 2021), a LM pretrained on diverse languages, but isn't instruction tuned or CoT fine-tuned, (3) MT0 (Muennighoff et al., 2022), a LM that is instruction-tuned on diverse languages, but isn't CoT fine-tuned. In relatively underrepresented languages such as Korean, Japanese, and Chinese, all three LMs get close to zero accuracy.

A natural question arises whether training a multilingual LM that could reason step-by-step on different languages is viable. As a preliminary research, we examine whether CoT Fine-tuning on a single language with a small amount of CoT data could enable LMs to avoid achieving near zero score such as Korean, Chinese and Japanese subsets of MGSM. Since there is no publicly available multilingual instruction dataset, we translate 60K ∼ 80K instances from CoT COLLECTION for each 5 languages using ChatGPT (OpenAI, 2022), and CoT fine-tune mT5 and mT0 on each of them.

The results are shown in Table 4. Across all the 5 different languages, CoT fine-tuning brings

| Method | ko | ru | fr | zh | ja |
|---|---|---|---|---|---|
| FLAN-T5-3B | 0.0 | 2.8 | 7.2 | 0.0 | 0.0 |
| FLAN-T5-11B | 0.0 | 5.2 | 13.2 | 0.0 | 0.0 |
| MT5-3.7B | 0.0 | 1.2 | 2.0 | 0.8 | 0.8 |
| MT0-3.7B | 0.0 | 4.8 | 7.2 | 1.6 | 2.4 |
| GPT-3 (175B) | 0.0 | 4.4 | 10.8 | 6.8 | 0.8 |
| MT5-3.7B + CoT FT | 3.2 | 6.8 | 9.6 | 6.0 | 7.6 |
| MT0-3.7B + CoT FT | 7.6 | 10.4 | 15.6 | 11.2 | 11.0 |

Table 4: Evaluation performance on MGSM benchmark (Shi et al., 2022) across 5 languages (Korean, Russian, French, Chinese, Japanese, respectively). All evaluations are held in a zero-shot setting with CoT Evaluation except GPT-3 using a 6-Shot prompt for ICL. The best comparable performances are **bolded** and second best underlined. Note that 'MT5-3.7B + CoT FT' and 'MT0-3.7B + CoT FT' are trained on a single language instead of multiple languages as mT5 and mT0.

about non-trivial gains in performance. Even for relatively low-resource languages such as Korean Japanese, and Chinese, CoT fine-tuning on the specific language allows the underlying LM to perform mathematical reasoning in the target language, which are considered very difficult (Shi et al., 2022). Considering that only a very small number of instances were used for language-specific adaptation (60k-80k), CoT fine-tuning shows potential for efficient language adaptation.

However, it is noteworthy that we limited our setting to training/evaluating on a single target language, without exploring the cross-lingual transfer of CoT capabilities among varied languages. The chief objective of this experimentation was to ascertain if introducing a minimal volume of CoT data could facilitate effective adaptation to the target language, specifically when addressing reasoning challenges. Up to date, no hypothesis has suggested that training with CoT in various languages could enable cross-lingual transfer of CoT abilities among different languages. We identify this as a promising avenue for future exploration.

### 4.3 Few-shot Generalization

In this subsection, we show how CoT-T5 performs in a few-shot adaptation setting where a limited number of instances from the target task can be used for training, which is sometimes more likely in real-world scenarios.

**Dataset Setup** We choose 4 domain-specific datasets from legal and medical domains including LEDGAR (Tuggener et al., 2020), Case Hold (Zheng et al., 2021), MedNLI (Romanov and Shivade, 2018), and PubMedQA (Jin et al., 2019). To simulate a few-shot setting, we randomly sample 64 instances from the train split of each dataset.

| Method | #Train Param | Ledgar | Case Hold | MedNLI | PubmedQA | Total Avg |
|---|---|---|---|---|---|---|
| Flan-T5-3B + Full FT. | 2.8B | 52.60 | 61.40 | 66.82 | 66.28 | 61.78 |
| Flan-T5-3B + Full CoT FT. | 2.8B | 53.60 | 58.80 | 65.89 | 65.89 | 61.05 |
| CoT-T5-3B + Full CoT FT. (Ours) | 2.8B | 51.90 | 60.60 | 67.16 | 68.12 | 61.95 |
| Flan-T5-3B + LoRA FT. | 2.35M | 53.20 | 58.80 | 61.60 | 67.18 | 60.19 |
| Flan-T5-3B + LoRA CoT FT. | 2.35M | 51.20 | 61.60 | 62.59 | 66.06 | 60.36 |
| CoT-T5-3B + LoRA CoT FT. (Ours) | 2.35M | 54.80 | 63.60 | 68.00 | 69.66 | 64.02 |
| Flan-T5-11B + LoRA FT. | 4.72M | 55.30 | 64.90 | 75.91 | 70.25 | 66.59 |
| Flan-T5-11B + LoRA CoT FT. | 4.72M | 52.10 | 65.50 | 71.63 | 71.60 | 65.21 |
| CoT-T5-11B + LoRA CoT FT. (Ours) | 4.72M | **56.10** | **68.30** | **78.02** | **73.42** | **68.96** |
| Claude (Anthropic, 2023) + ICL | 0 | 55.70 | 57.20 | 75.94 | 54.58 | 60.85 |
| Claude (Anthropic, 2023) + CoT PT. | 0 | 34.80 | 43.60 | 76.51 | 52.06 | 51.74 |
| ChatGPT (OpenAI, 2022) + ICL | 0 | 51.70 | 32.10 | 70.53 | 65.59 | 54.98 |
| ChatGPT (OpenAI, 2022) + CoT PT. | 0 | 51.00 | 18.90 | 63.71 | 25.22 | 39.70 |

Table 5: Evaluation performance on 4 domain-specific datasets. FT. denotes Fine-tuning, CoT FT. denotes CoT fine-tuning, and CoT PT. denotes CoT Prompting. The best comparable performances are **bolded** and second best underlined. For a few-shot adaptation, we use 64 randomly sampled instances from each dataset.

We report the average accuracy across 3 runs with different random seeds. We augment rationales for the 64 training instances using the procedure described in Section 3 for the rationale augmentation phase, utilizing the MCQA prompt from P3 dataset. In an applied setting, practitioners could obtain rationales written by human experts.

**Training Setup** We compare Flan-T5 & CoT-T5, across 3B and 11B scale and explore 4 different approaches for few-shot adaptation: (1) regular fine-tuning, (2) CoT fine-tuning, (3) LoRA fine-tuning, and (4) LoRA CoT fine-tuning. When applying Lora, we use a rank of 4 and train for 1K steps following Liu et al. (2022b). This results in training 2.35M parameters for 3B scale models and 4.72M parameters for 11B scale models. Also, we include Claude (Anthropic, 2023) and ChatGPT (OpenAI, 2022) as ICL baselines by appending demonstrations up to maximum context length[6]. Specifically, For CoT prompting, the demonstrations are sampled among 64 augmented rationales are used.

**Effect of LoRA** The experimental results are shown in Table 5. Overall, CoT fine-tuning CoT-T5 integrated with LoRA obtains the best results overall. Surprisingly for Flan-T5, applying full fine-tuning obtains better performance compared to its counterpart using LoRA fine-tuning. However, when using CoT-T5, LoRA achieves higher performance compared to full fine-tuning. We conjecture this to be the case because introducing only

a few parameters enables CoT-T5 to maintain the CoT ability acquired during CoT fine-tuning.

**Fine-tuning vs. CoT Fine-tuning** While CoT fine-tuning obtains similar or lower performance compared to regular fine-tuning in Flan-T5, CoT-T5 achieves higher performance with CoT fine-tuning compared to Flan-T5 regular fine-tuning. This results in CoT-T5 in combination with CoT fine-tuning showing the best performance in few-shot adaptation setting.

**Fine-tuning vs. ICL** Lastly, fine-tuning methods obtain overall better results compared to ICL methods utilizing much larger, proprietary LLMs. We conjecture this to be the case due to the long input length of legal and medical datasets, making appending all available demonstrations (64) impossible. While increasing the context length could serve as a temporary solution, it would still mean that the inference time will increase quadratically in proportion to the input length, which makes ICL computationally expensive.

## 5 Analysis of of CoT Fine-tuning

In this section, we conduct experiments to address the following two research questions:

- For practitioners, is it more effective to augment CoT rationales across diverse tasks or more instances with a fixed number of tasks?

- During CoT fine-tuning, does the LM maintain its performance on in-domain tasks without any catastrophic forgetting?

---

[6]Full context length was 4k tokens for ChatGPT and 9k tokens for Claude.

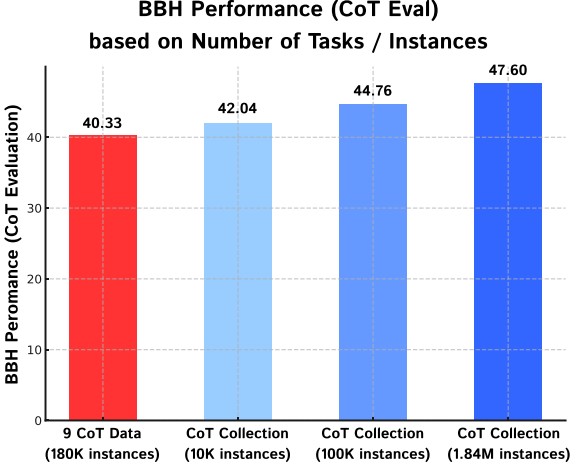

Figure 3: Scaling plot of increasing the number of instances within the CoT COLLECTION compared to using the existing 9 CoT datasets. Even with less number of instances, maintaining a wider range of tasks is crucial to improve the CoT abilities of an underlying LLM.

## 5.1 Scaling the number of tasks & instances

In our main experiments, we used a large number of instances (1.84M) across a large number of tasks (1,060) to apply CoT fine-tuning. A natural question arises: "Is it more effective to increase the number of tasks or the number of instances?" To address this question, we conduct an experiment of randomly sampling a small number of instances within the CoT COLLECTION and comparing the BBH performance with (1) a baseline that is only CoT fine-tuned with the existing 9 CoT tasks and (2) CoT-T5 that fully utilizes all the 1.84M instances. Specifically, we sample 10K, 100K instances within the CoT COLLECTION and for the 9 CoT tasks, we fully use all the 180K instances. As CoT-T5, we use Flan-T5 as our base model and use the same training configuration and evaluation setting (CoT Eval) during our experiments.

The results are shown in Figure 3, where surprisingly, only using 10K instances across 1,060 tasks obtains better performance compared to using 180K instances across 9 tasks. This shows that maintaining a wide range of tasks is more crucial compared to increasing the number of instances.

## 5.2 In-domain Task Accuracy of CoT-T5

It is well known that LMs that are fine-tuned on a wide range of tasks suffer from catastrophic forgetting (Chen et al., 2020; Jang et al., 2021, 2023), a phenomenon where an LM improves its performance on newly learned tasks while the performance on previously learned tasks diminishes.

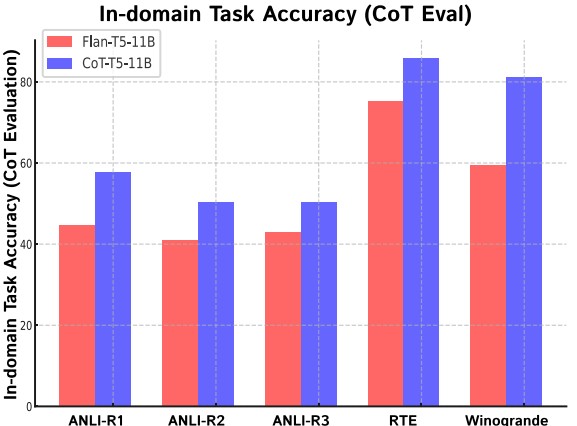

Figure 4: In-domain task accuracy with CoT evaluation. CoT Fine-tuning with the CoT COLLECTION also improves accuracy on in-domain tasks as well.

While CoT-T5 uses the same tasks as its base model (Flan-T5), we also check whether CoT fine-tuning on a wide range of tasks could possibly harm performance. For this purpose, we use the test set of 5 tasks within the CoT COLLECTION, namely ANLI-R1, ANLI-R2, ANLI-R3, RTE, and Winogrande. Note that this differs with the Setup #2 in the main experiments in that we use different base models (T0 vs Flan-T5), and the tasks are already used for CoT fine-tuning.

Results are shown in Figure 4, where CoT-T5 consistently improves in-domain accuracy on the learned tasks as well. However, we conjecture that this is because we used the exact same task that Flan-T5 used to CoT fine-tuned CoT-T5. Adding additional tasks that were not used to train Flan-T5 and CoT-T5 could show different results, and we leave additional exploration of catastrophic forgetting during CoT fine-tuning to future work.

## 6 Conclusion

In this work, we show that augmenting rationales from an instruction tuning data using LLMs (Open AI Codex), and CoT fine-tuning could improve the reasoning capabilities of smaller LMs. Specifically, we construct CoT COLLECTION, a large-scale instruction-tuning dataset with 1.84M CoT rationales extracted across 1,060 NLP tasks. With our dataset, we CoT fine-tune Flan-T5 and obtain CoT-T5, which shows better zero-shot generalization performance and serves as a better base model when training with few number of instances. We hope CoT COLLECTION could be beneficial in the development of future strategies for advancing the capabilities of LMs with CoT fine-tuning.

## Acknowledgments

This work was partly supported by Institute of Information & communications Technology Planning & Evaluation (IITP) grant funded by the Korea government (MSIT) (No.2022-0-00264, Comprehensive Video Understanding and Generation with Knowledge-based Deep Logic Neural Network, 50%; No.2021-0-02068, Artificial Intelligence Innovation Hub, 20%) and KAIST-NAVER Hypercreative AI Center.

## Limitations

Recently, there has been a lot of focus on distilling the ability to engage in dialogues with long-form outputs in the context of instruction following (Taori et al., 2023; Chiang et al., 2023). Since our model CoT-T5 is not trained to engage in dialogues with long-form responses from LLMs, it does not necessarily possess the ability to be applied in chat applications. In contrast, our work focuses on improving the zero-shot and few-shot capabilities by training on academic benchmarks (CoT COLLECTION, Flan Collection), where LMs trained with chat data lack on. Utilizing both long-form chat data from LLMs along with instruction data from academic tasks has been addressed in future work (Wang et al., 2023). Moreover, various applications have been introduced by using the FEEDBACK COLLECTION to train advanced chat models [7].

Also, since CoT-T5 uses Flan-T5 as a base model, it doesn't have the ability to perform step-by-step reasoning in diverse languages. Exploring how to efficiently and effectively train on CoT data from multiple languages is also a promising and important line of future work. While Shi et al. (2022) has shown that large LMs with more than 100B parameters have the ability to write CoT in different languages, our results show that smaller LMs show nearly zero accuracy when solving math problems in different languages. While CoT fine-tuning somehow shows slight improvement, a more comprehensive strategy of integrating the ability to write CoT in diverse language would hold crucial.

In terms of reproducibility, it is extremely concerning that proprietary LLMs shut down such as the example of the Codex, the LLM we used for rationale augmentation. We provide additional analysis on how different LLMs could be used for this

[7] https://huggingface.co/aiplanet/effi-13b

process in Appendix A. Also, there is room of improvement regarding the quality of our dataset by using more powerful LLMs such as GPT-4 and better prompting techniques such as Tree of Thoughts (ToT) (Yao et al., 2023). This was examined by later work in Mukherjee et al. (2023) which used GPT-4 to augment 5 million rationales and Yue et al. (2023) which mixed Chain-of-Thoughts and Program of Thoughts (PoT) during fine-tuning. Using rationales extracted using Tree of Thoughts (Yao et al., 2023) could also be explored in future work.

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

# A Analysis of CoT COLLECTION

Non-cherry picked rationales within CoT COLLECTION are shown in Table 8. We perform an analysis regarding the quality, diversity, and reproducibility of rationale within the CoT COLLECTION.

**Diversity of Rationales** To take a look into the diversity of CoT COLLECTION, we use Berkeley Neural Parser (Kitaev and Klein, 2018; Kitaev et al., 2019) and parse rationales. More specifically, the verb which is closest to the root of the parse tree along the noun object is extracted. We compare this with the rationales from the 9 CoT datasets used in Chung et al. (2022). As shown in Figure 5, CoT COLLECTION have diverse textual formats included compared to the 9 existing CoT datasets that have a high proportion assigned to 'answer question' and 'consider following'.

**Quality of Rationales** To ensure the quality of CoT COLLECTION, we use ROSCOE (Golovneva et al., 2022), a suite of metrics designed to evaluate rationales under different criteria within semantic alignment, semantic similarity, logical inference, language coherence. We compare with human-authored rationales obtained during Prompt Creation in Section 3. The 13 ROSCOE scores are shown in Table 6. The results show that CoT COLLECTION include CoT rationales that are faithful, less repetitive, informative, and logical even when compared to human-authored rationales. Yet, we find that machine-generated rationales tend to have higher perplexity, leading to lower language coherence scores. We conjecture this is because including diverse textual formats leads may result in relatively higher perplexity (Holtzman et al., 2019).

**Is CoT COLLECTION Reproducible?** One could doubt whether CoT COLLECTION is reproducible due to the usage of OpenAI model in the process of CoT rationale augmentation[8]. In this section, we test different LLMs to generate 150 rationales randomly sampled from CoT COLLECTION, and compare the ROSCOE score (Golovneva et al., 2022) in order to assess the quality. We use Bard (Google, 2023), Claude (Anthropic, 2023), for comparing with OpenAI Codex. The comparison of quality is shown in Figure 6. The results

---

[8]Moreover, OpenAI announced to stop its support on Codex model starting from June, 2023.

| | Metrics | Human | CoT Collection |
|---|---|---|---|
| Semantic Alignment | faithfulness | 0.8836 | **0.8914** |
| | faithfulness_ww | 0.8756 | **0.8793** |
| | repetition_word | 0.9376 | **0.9419** |
| | informativeness_step | 0.9519 | **0.9521** |
| Semantic Similarity | informativeness_chain | 0.2295 | **0.2797** |
| | repetition_sent | 0.2453 | **0.2910** |
| Logical Inference | discourse_representation | **0.4855** | 0.4687 |
| | coherence_step_vs_step | 0.7763 | **0.7813** |
| Language Coherence | perplexity_step | **0.0198** | 0.0122 |
| | perplexity_chain | **0.0475** | 0.0255 |
| | perplexity_step_max | **0.0144** | 0.0088 |
| | grammar_step | **0.8883** | 0.8721 |
| | grammar_step_max | **0.8013** | 0.7724 |

Table 6: Comparison of the quality between human-authored rationales and machine-generated rationales. 13 label-free metrics from ROSCOE (Golovneva et al., 2022) is used.

show that different LLMs are able to produce high quality rationales in terms of semantic alignment and language coherence.

# B Filtering CoT COLLECTION

**Filtering** After generating multiple rationales, we filter to ensure high quality. We apply the following criteria to filter instances:

- We exclude rationales that do not include the ground truth answer when splitted by white spaces. While a rationale that doesn't include the answer isn't necessarily a bad rationale, we found it is effective to exclude inconsistent ones.

- We exclude CoT rationales that exceed the maximum output length, where we constrain the sum of $r$ and $y$ to be shorter than 512 tokens.

- We exclude rationales that are identical to previously augmented ones during our process.

- We exclude rationales that include repetitive sentences within the context.

We further include the filtered instances in Table 9.

Also, we found that in many cases, Codex degenerates and starts writing code after the rationale. To prevent inclusion of code snippets, we apply additional filtering based on trigger tokens that abundantly appear in the start of the code. The list of trigger tokens are as follows:

```
CODE_FILTER = [
    "\n`\n\n'''", "\n`\n", "\n''\n",
    "\n'''\n", "\n```", "\n\n \n'",
    "\n\n \n`", "\n\n \nimport",
    "\n`;\n\n", "\"\n\n",
    "[examp", "[Examp", "\n`;\n\n",
    "'''\n", "\n`\n", "\n\n``",
    "\n``", "\"\"\"\n", "\n\n\t",
    "\n#", "\";\n", "\"\n\t",
    "print(", "\n     ", "\"\n'''",
    "'''\nimport", "\"\n\n\t",
    "\n\n    ", "\n\t\t", "\t    ",
    "\"\n }\n", "\n\n ####",
    "\n\n \t')\n}", "\n</block>\n\n",
    "\n\n      */\n",
    "\"\n\n \n \t`;", "\n\n \t",
    "\"\n\n \t */", "\";\n\n }",
    "\n\n \t\t", "\"\n`,\n\t}",
    "]]\n\t", "\"\n\n`", "\"\n'''\n\n",
    "\n\n         OR", "\n \n"
]
```

## C   Training and Evaluation Details of CoT-T5

| Params | Model | Batch size | LR | Optimizer |
|--------|-------|-----------|------|-----------|
| 3B | CoT-T5-3B | 64 | 5e-5 | AdamW |
| 11B | CoT-T5-11B | 8 | 1e-4 | Adafactor |

Table 7: Hyperparameters used for fine-tuning CoT-T5.

We mostly follow the fine-tuning details of Chung et al. (2022) to train CoT-T5. The hyperparameters used for training CoT-T5 are shown in Table 7. We find 3B and 11B sized LMs converge well using different optimizers. While CoT-T5-3B tends to converge well using AdamW, CoT-T5-11B is well optimized using Adafactor. For both sizes, we train with 1 epoch using CoT COLLECTION which takes 1 day (3B) and 7 days (11B) when 8 A 100 (80GB) GPUs are used. For both settings, we use a gradient accumulation step of 8.

For sampling training instances, we sample instances from Flan Collection (Longpre et al., 2023) by using the proportion of 23.94%(FLAN), 30.85%(P3), 7.89%(Existing 9 CoT datasets), 25.47%(SNI) and 11.85%(other dialogue & code datasets). This is done by sampling 400 instances (FLAN), 300 instances (P3), 150 instances (SNI), 4000 instances (Existing 9 CoT datasets), and 300 instances (other dialogue & code datasets), respectively. We generate 5 rationales per instance and then apply filtering, leading to the final set of CoT COLLECTION, which is consisted of 1.84 million instances and rationales across 1,060 tasks.

During evaluation, we found that using nucleus sampling (Holtzman et al., 2019) with $p = 0.8$ and no_repeat_n_gram $= 3$ was very effective in generating good-quality rationales.

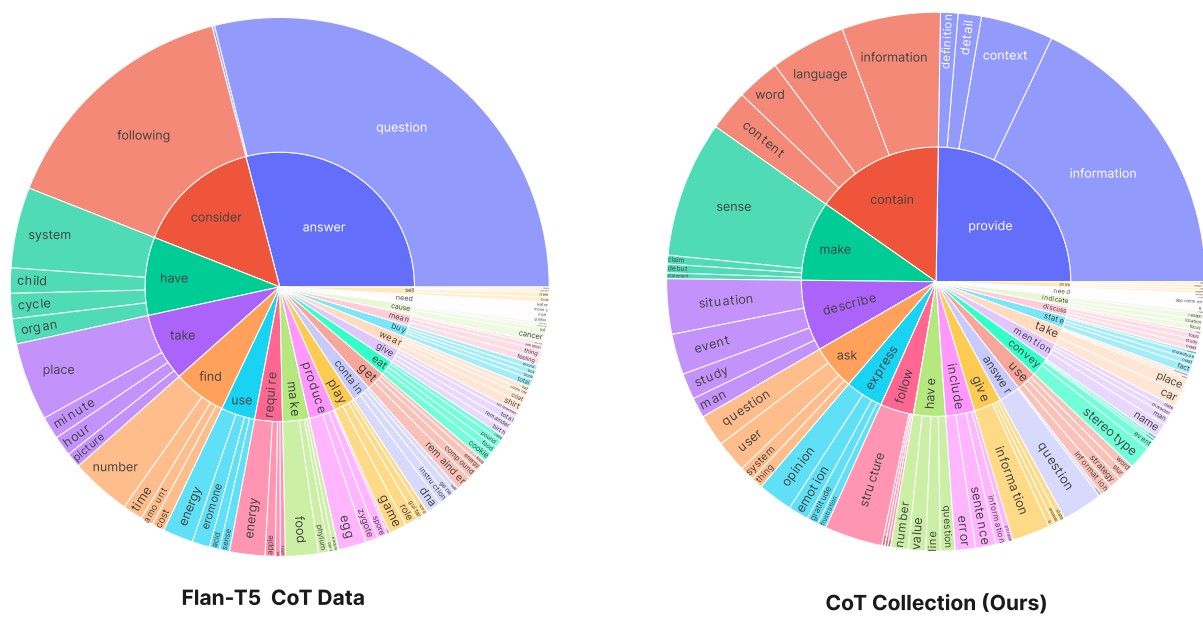

Figure 5: The top 20 common root verbs (inner circle) and their top 4 noun objects (outer circle) within the rationales of the 9 CoT tasks used in Chung et al. (2022) (left side) and our CoT COLLECTION with 1,060 tasks (right side).

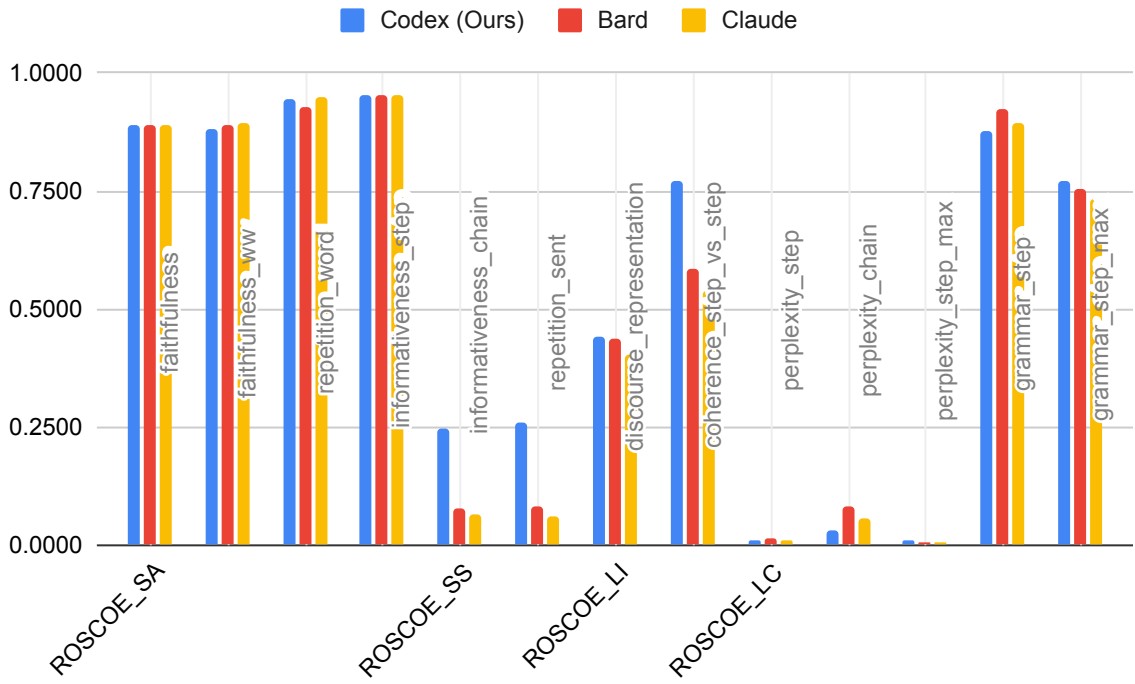

Figure 6: Comparison of the quality of augmenting CoT rationales using different LLMs by reference-free ROSCOE scores. Semantic Alignment(SA), Semantic Similarity(SS), Logical Inference(LI), Coherence(LC).

| Question & Instruction | Answer | Rationale |
|---|---|---|
| Read the following context and choose the best option to answer the question.

Context: From the triple loop she executed after five minutes, he suspected that she had managed to sneak off for flying more often than during the summers and wondered just how diligent George was about practicing on his own time.

Question: What is a triple loop ?

Options:
A. None of the above choices .
B. 3 round trips on a plane route .
C. 3 neckties worn on flights
D. A feature pulled off by making 3 loops in the air. | D | A triple loop is a maneuver in which an aircraft makes three loops. Based on the context, it can be inferred that a triple loop involves making 3 loops in the air. Since this is what A means, D should be chosen as the correct answer. |
| Read the passage below and choose the right answer to the following question (choices are more or less ):

On a hot day, an endotherm may give off more heat by increasing blood flow to the surface of the body.

After Sarah finishes her walk, she notices that her blood is not really pumping that hard. Sarah can know that since it's a cool day, she's giving off what kind of heat compared to usual more or less? | less | The given passage states that an endotherm gives off more heat by increasing blood flow to the surface of its body. As such, it can be inferred that decreased blood flow corresponds with less heat being given off. Since Sarah notices that her blood is not really pumping hard on a cool day, she is giving off less heat compared to usual. |
| Read this and answer the question.
Pubertal development also affects circulatory and respiratory systems as an adolescents' heart and lungs increase in both size and capacity. These changes lead to increased strength and tolerance for exercise. Sex differences are apparent as males tend to develop "larger hearts and lungs, higher systolic blood pressure, a lower resting heart rate, a greater capacity for carrying oxygen to the blood, a greater power for neutralizing the chemical products of muscular exercise, higher blood hemoglobin and more red blood cells.

Which sex tends to develop larger hearts and lungs? | males | The last sentence of the paragraph states that males tend to develop larger hearts and lungs. |
| In this task, you are given a string of characters. You need to remove duplicate characters from the string if any, and return the resulting string.

"nXnGFzHFYudOiHyWLZXQfpw" | nXGFzHYudOiyWLZQfpw | There are no duplicate characters in the given string, so the resulting string will be the same as the input. Therefore, the output will be "nXGFzHYudOiyWLZQfpw". |
| In this task you will be given a list of numbers and you should remove all duplicates in the list. If every number is repeated in the list an empty list should be returned. Your list should be numbers inside brackets, just like the given list.

{3, 5, 1, 0, 3} | {5, 1, 0} | The given list is {3, 5, 1, 0, 3}. We can see that '3' appears in the list twice. Therefore, we need to keep only one of the '3's and remove the other one. The resulting list is {5, 1, 0}. |
| How to keep your phone cool in the car
Keep your phone on a vent mount with air conditioning on if you need to use it. If you need to use your phone while you're in your car, purchase a mount that attaches to one of the dashboard air vents. Turn on the air conditioner so it keeps your phone cold.

Can we draw the following conclusion?
keep your phone and tablet cool while mounted on an air conditioner vent if you need to use them while in the car.

OPTIONS:
- Yes
- It's impossible to say
- No | It's impossible to say | While the passage does mention that your phone should be mounted on an air vent, it doesn't specify whether this is referring to a car or home AC. Additionally, we don't know from the context if other devices such as tablets can also be kept cool using this method.

So, "It's impossible to say" |
| Give me a possible correct answer to the question "who nominated stephen g. breyer?" | Bill Clinton | Stephen G. Breyer was a nominee for the Supreme Court by Bill Clinton and was confirmed on July 29, 1994. Although he wasn't nominated in 1998 or 1999, this is still a valid answer because it happened within 5 years of those dates before. |
| Write down the solution for this math problem:
Solve $291*c - 264*c = 189$ for c. | 7 | STEP 1. $291*c - 264*c = 189$
STEP 2. $27*c = 189$
STEP 3. $c = 7$ |

Table 8: Example of rationales within CoT COLLECTION including tasks such as Multiple choice QA (MCQA), Extractive QA (ExQA), Closed-book QA (CBQA), Formal Logic, Natural Language Inference (NLI), and Arithmetic.

| Question | Answer | CoT Rationale |
|---|---|---|
| You are given a math word problem and you are supposed to only use subtraction on the numbers embedded in the text to answer the following question and then only report the final numerical answer.

Context: Jake has 8 fewer peaches and 10 more apples than Steven. Steven has 11 apples and 18 peaches.

Question: How many more peaches than apples does Steven have? | 7 | To solve this problem,
we need to find the number of apples and peaches that Steven has.
From the text, we know that Jake has 8 fewer peaches
than Steven and 10 more apples than Steven.
We also know that Steven has
11 apples and 18 peaches.
Using these clues we can write two equations:
Steven's peaches = Jake's Peaches + 8
Steven's Apples = Jake's Apples - 10
We can then add these two equations together:
(Steven's Peaches)+(Stevens' Apples) = (Jake's Peaches+8) + (Jake's Apples-10)
(18+11)=(Jake 's Peaches +8)+ (Jakes' Apples-10)
29= Jakes'peach+Jakes'applees-2
a=29-2=27
b=(18+11)-a
c=b/2
nd=c*2 |
| What is the solution?
Solve 690*c + 14495 = -4825 for c. | -28 | 690*c + 14495 = -4825
Subtracting 4825 from both sides:
690*c + 9669 = 0
Dividing both sides by 690:
c=-14.02391304348 |

Table 9: Two representative rationale examples that are filtered with our filtering criteria. It demonstrates that not including the ground truth token within the rationale could indicate that the LLM failed to generate a consistent rationale, often leading to degeneration. This phenomenon is mostly found on math datasets.

# D   Prompts used for Rationale Augmentation

Among the 26 demonstrations used for each family of tasks when augmenting CoT rationales, we include 4 examples, other demonstrations for task families can be seen at this link.

---

[Example 1]
[Instruction and Question]
Skylar had stopped breathing but Lee held Skylar's arms anyway, in vain. Given the context: Lee want to do what next?
Possible answers: beg the doctors to try again, get her up and walking home, bring Skylar back

[Options]
– beg the doctors to try again
– get her up and walking home
– bring Skylar back

[Answer]
beg the doctors to try again

[Rationale]
The context of the situation is that Skylar has stopped breathing and Lee is holding Skylar's arms. This suggests that Skylar has received medical attention and is no longer alive.
Based on this, Lee would want to beg the doctors to try again, as this would be the best way to potentially save Skylar's life.
The options of getting Skylar up and walking home or bringing Skylar back do not align with the context, as they would not address the immediate medical emergency at hand.

So the answer is to beg the doctors to try again.

[Example 2]
[Instruction and Question]
Do you think the right answer to the question "what can run on alcoholic fermentation of the glucose in corn or other plants?" is "bicycles"?

[Options]
– No
– Yes

[Answer]
No

[Rationale]
Alcoholic fermentation is a process that occurs in living organisms, most commonly yeast, to convert glucose into energy. This process does not produce enough energy to power something like a bicycle.

So the answer is No.

[Example 3]
[Instruction and Question]
Eventually , the cyst can get so large that part of the wall weakens and fluid begins to leak out ; that is , the cyst ruptures . It is similar to popping a water balloon . When this happens , the fluid may irritate the pelvic lining and cause pain , often severe . Most of the time the pain begins to gradually subside over the course of the next few days .
What type of doctor might the person have had to see ?

– A paediatrician – A heart surgeon – a gynecologist – None of the above choices

[Options]
– A paediatrician
– A heart surgeon
– a gynecologist
– None of the above choices

[Answer]
a gynecologist

[Rationale]
The context mentions a cyst in the pelvic region and the symptoms of the cyst rupturing, including pain and fluid leakage. This suggests that the issue is related to the reproductive or urinary system. Given that the issue is related to the reproductive or urinary system, it is likely that the person would have had to see a specialist in these areas.
Out of the given options, "a gynecologist" is the most likely specialist that the person would have had to see, as gynecologists specialize in female reproductive health, including issues related to the uterus, ovaries, and other reproductive organs.

In conclusion, based on the given context, the most likely answer is a gynecologist.

[Example 4]
[Instruction and Question]
Process:
– Get some seeds

– Pick a spot to plant them
– Dig a hole in the dirt
– Put the seed in the hole
– Pour some water on the seed and hole
– Cover up the hole
– Press down on it
– Spray some plant food on it.

Question:
suppose if no water is available happens, how will it affect vegetables?

– A: more
– B: less

[Options]
– A
– B

[Answer]
B

[Rationale]
The process of planting seeds requires water, which is essential for the growth of plants. Water helps transport nutrients, regulate temperature, and absorb minerals from the soil. If no water is available, the seed will not germinate and the plant will not grow properly, leading to fewer vegetables. Hence, the absence of water will result in "less" vegetables, making the answer "B".

[Example 5]
[Instruction and Question]
Read the following conversation and answer the question.
M: I'm worried about my sister, Lucy.
W: Why is that?
M: She thinks she's too fat.
W: Is she?
M: No, but she keeps skipping meals. Then, she only eats chips and drinks cola.
W: I used to do that. It's called binge. It was no fun!
M: Why did you stop doing it?
W: Well, my doctor told me to eat when I'm hungry. She said, "Eat till you're full or you'll eat too much later." She said a lot of girls ruin their health this way.
M: Did she say what to eat?
W: She said, "Eat fruit, vegetables, meats and grains. Have regular meals and snacks. Get exercise, too."

Question: What's the relationship between the speakers?

– Doctor and patient.
– Friends.
– Brother and sister.

[Options]
– Doctor and patient
– Friends
– Brother and sister

[Answer]
Friends

[Rationale]
The conversation between two people, a man(M) and a woman(W), is discussing the man's sister who has a negative view of her own body and is skipping meals. The woman shares that she used to engage in similar behavior, and explains that her doctor advised her to eat a balanced diet and exercise.

Among the three options,
– First, since the woman is sharing her experience of what the doctor told her, it is unlikely she would be a medical professional. So the answer couldn't be doctor and patient.
– Second, given that the man is talking about his sister, the woman is unlikely to be the man's sister.
– Lastly, because the conversation is informal and the speakers are discussing a personal matter it is likely that the relationship between the speakers is that of friends.

Therefore, the answer is Friends.

[Example 6]
[Instruction and Question]
Question: The best place for the troublemaker to sneak a smoke while he was in where, was in the bathroom?

Choices:
– playground
– school
– at hotel
– theater
– convenience store

The rationale to choose ""school"" as the answer is that:

[Options]
Not Given

[Answer]
smokers traditionally use school bathrooms

[Rationale]
Here's the rationale: "smokers traditionally use school bathrooms".

[Example 7]
[Instruction and Question]
So I was looking at quotes for our tshirts for ice hockey , and I came across one that sums up how I ' m feeling right now . When you talk about an injury and the kind of depression you go through , it 's not just because you 're out of shape and you can't go out and play . You 're missing a part of you .

Why is the narrator so down in the dumps ?
Pick the best answer from the following options:
A. They are unable to play ice curling .
B. They are hurt .
C. They are unable to play ice skating .
D. They are unable to play in the yard .

[Options]
– A
– B
– C
– D

[Answer]
B

[Rationale]
The quote in the question mentions the narrator's feelings about an injury and the depression that follows. The narrator says that it's not just about being out of shape and unable to play, but rather they feel like they are missing a part of themselves. This implies that the reason for their depression is linked to their physical inability to participate in a sport or activity they enjoy. Based on the context, it can be inferred that the narrator is upset because they are hurt and unable to play ice hockey.
So, the answer is B.

[Example 8]
[Instruction and Question]
Q: What does the ubiquitous generator use to generate currents?

Choices:

– inertia – water – gravity – magnetism

[Options]
– inertia
– water
– gravity
– magnetism

[Answer]
magnetism

[Rationale]
A generator is a device that converts mechanical energy into electrical energy. This is done by using a magnetic field to induce a current in the generator's windings. This current is then sent to an external circuit, where it can be used to power electrical devices. The generator uses magnetism to generate currents, so the answer is magnetism.

[Example 9]
[Instruction and Question]

# Prompts 1: Demonstration used for tasks in family of MCQA (T0)

[Example 1]
[Instruction and Question]
Skylar had stopped breathing but Lee held Skylar's arms anyway, in vain. Given the context: Lee want to do what next?
Possible answers: beg the doctors to try again, get her up and walking home, bring Skylar back

[Options]
– beg the doctors to try again
– get her up and walking home
– bring Skylar back

[Answer]
beg the doctors to try again

[Rationale]
The context of the situation is that Skylar has stopped breathing and Lee is holding Skylar's arms. This suggests that Skylar has received medical attention and is no longer alive.
Based on this, Lee would want to beg the doctors to try again, as this would be the best way to potentially save Skylar's life.
The options of getting Skylar up and walking home or bringing Skylar back do not align with the context, as they would not address the immediate medical emergency at hand.

So the answer is to beg the doctors to try again.

[Example 2]
[Instruction and Question]
Do you think the right answer to the question "what can run on alcoholic fermentation of the glucose in corn or other plants?" is "bicycles"?

[Options]
– No
– Yes

[Answer]
No

[Rationale]
Alcoholic fermentation is a process that occurs in living organisms, most commonly yeast, to convert glucose into energy. This process does not produce enough energy to power something like a bicycle.

So the answer is No.

[Example 3]
[Instruction and Question]
Eventually , the cyst can get so large that part of the wall weakens and fluid begins to leak out ; that is , the cyst ruptures . It is similar to popping a water balloon . When this happens , the fluid may irritate the pelvic lining and cause pain , often severe . Most of the time the pain begins to gradually subside over the course of the next few days .
What type of doctor might the person have had to see ?

– A paediatrician – A heart surgeon – a gynecologist – None of the above choices

[Options]
– A paediatrician
– A heart surgeon
– a gynecologist
– None of the above choices

[Answer]
a gynecologist

[Rationale]
The context mentions a cyst in the pelvic region and the symptoms of the cyst rupturing, including pain and fluid leakage. This suggests that the issue is related to the reproductive or urinary system. Given that the issue is related to the reproductive or urinary system, it is likely that the person would have had to see a specialist in these areas.
Out of the given options, "a gynecologist" is the most likely specialist that the person would have had to see, as gynecologists specialize in female reproductive health, including issues related to the uterus, ovaries, and other reproductive organs.

In conclusion, based on the given context, the most likely answer is a gynecologist.

[Example 4]
[Instruction and Question]
Process:
– Get some seeds
– Pick a spot to plant them
– Dig a hole in the dirt
– Put the seed in the hole
– Pour some water on the seed and hole
– Cover up the hole
– Press down on it
– Spray some plant food on it.

Question:
suppose if no water is available happens, how will it affect vegetables?

– A: more
– B: less

[Options]
– A
– B

[Answer]
B

[Rationale]
The process of planting seeds requires water, which is essential for the growth of plants. Water helps transport nutrients, regulate temperature, and absorb minerals from the soil. If no water is available, the seed will not germinate and the plant will not grow properly, leading to fewer vegetables. Hence, the absence of water will result in "less" vegetables, making the answer "B".

[Example 5]
[Instruction and Question]
Read the following conversation and answer the question.
M: I'm worried about my sister, Lucy.
W: Why is that?
M: She thinks she's too fat.
W: Is she?
M: No, but she keeps skipping meals. Then, she only eats chips and drinks cola.

W: I used to do that. It's called binge. It was no fun!

M: Why did you stop doing it?

W: Well, my doctor told me to eat when I'm hungry. She said, "Eat till you're full or you'll eat too much later." She said a lot of girls ruin their health this way.

M: Did she say what to eat?

W: She said, "Eat fruit, vegetables, meats and grains. Have regular meals and snacks. Get exercise, too."

Question: What's the relationship between the speakers?

– Doctor and patient.
– Friends.
– Brother and sister.

[Options]
– Doctor and patient
– Friends
– Brother and sister

[Answer]
Friends

[Rationale]
The conversation between two people, a man(M) and a woman(W), is discussing the man's sister who has a negative view of her own body and is skipping meals. The woman shares that she used to engage in similar behavior, and explains that her doctor advised her to eat a balanced diet and exercise.

Among the three options,
– First, since the woman is sharing her experience of what the doctor told her, it is unlikely she would be a medical professional. So the answer couldn't be doctor and patient.
– Second, given that the man is talking about his sister, the woman is unlikely to be the man's sister.
– Lastly, because the conversation is informal and the speakers are discussing a personal matter it is likely that the relationship between the speakers is that of friends.

Therefore, the answer is Friends.

[Example 6]
[Instruction and Question]
Question: The best place for the troublemaker to sneak a smoke while he was in where, was in the bathroom?

Choices:
– playground
– school
– at hotel
– theater
– convenience store

The rationale to choose ""school"" as the answer is that:

[Options]
Not Given

[Answer]
smokers traditionally use school bathrooms

[Rationale]
Here's the rationale: "smokers traditionally use school bathrooms".

[Example 7]
[Instruction and Question]
So I was looking at quotes for our tshirts for ice hockey , and I came across one that sums up how I ' m feeling right now . When you talk about an injury and the kind of depression you go through , it 's not just because you 're out of shape and you can't go out and play . You 're missing a part of you .

Why is the narrator so down in the dumps ?
Pick the best answer from the following options:
A. They are unable to play ice curling .
B. They are hurt .
C. They are unable to play ice skating .
D. They are unable to play in the yard .

[Options]
– A
– B
– C
– D

[Answer]
B

[Rationale]
The quote in the question mentions the narrator's feelings about an injury and the depression that follows. The narrator says that it's not just about being out of shape and unable to play, but rather they feel like they are missing a part of themselves. This implies that the reason for their depression is linked to their physical inability to participate in a sport or activity they enjoy. Based on the context, it can be inferred that the narrator is upset because they are hurt and unable to play ice hockey.
So, the answer is B.

[Example 8]
[Instruction and Question]
Q: What does the ubiquitous generator use to generate currents?

Choices:

– inertia – water – gravity – magnetism

[Options]
– inertia
– water
– gravity
– magnetism

[Answer]
magnetism

[Rationale]
A generator is a device that converts mechanical energy into electrical energy. This is done by using a magnetic field to induce a current in the generator's windings. This current is then sent to an external circuit, where it can be used to power electrical devices. The generator uses magnetism to generate currents, so the answer is magnetism.

[Example 9]
[Instruction and Question]

## Prompts 2: Demonstration used for tasks in family of Commonsense (SNI)

[Example 1]
[Instruction and Question]
Problem: Context: Anorexia in males accounted for approximately six percent of cases seen in an eating disorder clinic.\nAnorexia becomes common in males.\nOPTIONS:\n– yes\n– no\nA:

[Answer]
no

[Rationale]
The information given only pertains to cases seen in this particular eating disorder clinic and may not be representative of the entire population. To determine whether anorexia is becoming common in males, we would need to look at data from a larger sample size or population, as well as data from multiple sources to verify the trend. So the answer is "no".

[Example 2]
[Instruction and Question]
Generate a context and a hypothesis.\n\nAnswer:

[Answer]
He also liked swimming and cycling. He said that he wrote stories, though he had to admit that he had never got further than the first two pages. Willie meanwhile not only remained silent during these conversations but picked his berries slowly so that they might forget that he was there but he reckoned without Zach.\n\nHypothesis: Willie was there

[Rationale]
Here's a context and an example for natural language inference task:\nHe also liked swimming and cycling. He said that he wrote stories, though he had to admit that he had never got further than the first two pages. Willie meanwhile not only remained silent during these conversations but picked his berries slowly so that they might forget that he was there but he reckoned without Zach.\n\nHypothesis: Willie was there

[Example 3]
[Instruction and Question]
Q: Brock was a popular English writer and director.\nJeremy Brock MBE (born 1959) is an English writer and director whose works include the screenplays "Mrs Brown", "Driving Lessons", "The Last King of Scotland", "Charlotte Gray", and "The Eagle". Brock has also written two plays for the Hampstead downstairs theatre.\nOPTIONS:\n– Yes\n– It's impossible to say\n– No\nA:

[Answer]
It's impossible to say

[Rationale]
While the information provided in the options about Jeremy Brock suggests that he is indeed an English writer and director with a number of successful works under his belt, there is not enough information to determine whether he is actually "popular" or not. The term "popular" is subjective and can vary depending on different factors, such as time period, genre, and audience. So the answer is "It's impossible to say".

[Example 4]
[Instruction and Question]
Biff Burger (Best In Fast Food) was a burger fast food franchise that originated in Clearwater, Florida. Early success in the late 1950s and early 1960s led the franchise to spread to many parts of the Southeastern United States. However, financial woes in the 1970s sent the chain on a decline and most of its restaurants out of business by the 1980s.

Today, two Biff Burger restaurants remain.\nBiff Burger came about in Clearwater Florida. OPTIONS:\n– Yes\n– It's impossible to say\n– No\nA:

[Answer]
Yes

[Rationale]
To validate the claim "Biff Burger came about in Clearwater Florida", we should find a supporting evidence within the context. We could see that "Biff Burger (Best In Fast Food) was a burger fast food franchise that originated in Clearwater, Florida." supports the claim we are trying to validate. So the answer is "Yes".

[Example 5]
[Instruction and Question]
input hypothesis: In the decade before the 1950s, the United States FBI began to maintain a public list of the people it regarded as the Ten Most Wanted Fugitives\nContext: In the 1950s, the United States FBI began to maintain a public list of the people it regarded as the Ten Most Wanted Fugitives. Following is a brief review of FBI people and events that place the 1950s decade in context, and then an historical list of individual fugitives whose names first appeared on the 10 Most Wanted list during the decade of the 1950s, under FBI Director J. Edgar Hoover.\nOPTIONS:\n– Yes\n– It's impossible to say\n– No\ntrue or false:

[Answer]
No

[Rationale]
The context provided contradicts this hypothesis by stating that the FBI began to maintain this public list in the 1950s, under the direction of J. Edgar Hoover. The context goes on to list individuals who were first placed on the 10 Most Wanted list during the 1950s, further supporting the notion that the list was not in existence prior to that decade. So the answer is "No".

[Example 6]
[Instruction and Question]
Q: If The plaintiffs claim Penrose's design is distinguished by its aperiodicity (the pattern almost but never quite repeats itself) and its five–fold symmetry (a trait that at the time was thought not to exist in nature but has since been identified in certain crystal formations)., can we say that "The design has no identified traits that would distinguish it."?\nOPTIONS:\n– yes\n– it is not possible to tell\n– no\n\nA:

[Answer]
no

[Rationale]
The plaintiffs are stating that the design has unique characteristics that set it apart, which directly contradicts the statement that the design has no identified traits that would distinguish it. Therefore, the answer is "no".

[Example 7]
[Instruction and Question]
Does "The first docking of Soviet spacecraft was finally realised in January 1969 by the Soyuz 4 and Soyuz 5 missions." contain the correct answer to "When was the first successful docking of a two man space crew?"\nOPTIONS:\n– yes\n– no\n\nA:

[Answer]
yes

[Rationale]
The passage talks about the first successful docking of a two–man space crew, as both Soyuz 4 and Soyuz 5 had two cosmonauts on board at the time of their docking. Therefore, the statement contains the correct answer to the question. So, the answer is "yes".

[Example 8]
[Instruction and Question]
input question: Is it true that "I shall have to guard The foxes." if "The foxes are getting in at night and attacking the chickens. I shall have to guard them." is true? OPTIONS:\n– no\n– yes\noutput answer:

[Answer]
no

[Rationale]
The statement "I shall have to guard The foxes" does not accurately reflect the situation described in the second sentence "I shall have to guard them". In this context, "them" refers to the foxes, so the correct statement would be "I shall have to guard them", not "I shall have to guard the foxes". So, the answer is "no".

[Example 9]
[Instruction and Question]
Question:\nPremise: A man in a blue shirt and orange shorts looks on at a small child wearing a yellow shirt and blue shorts with trees comprising the backdrop.\n\nHypothesis: There are a lot of trees behind the small child.\nIs the hypothesis true?\nOPTIONS:\n– yes\n– it is not possible to tell\n– no\n\nAnswer:\n

[Answer]
it is not possible to tell

[Rationale]
While it is mentioned that trees make up the backdrop, it is not specified whether there are a lot of trees or not. Additionally, the focus of the premise is on the man and child and not on the trees in the background. Hence, it is not possible to determine whether the hypothesis is true or not.

[Example 10]
[Instruction and Question]

---

## Prompts 3: Demonstration used for tasks in family of NLI (FLAN)

---

[Example 1]
[Instruction and Question]
There are 15 trees in the grove. Grove workers will plant trees in the grove today. After they are done, there will be 21 trees. How many trees did the grove workers plant today?

[Answer]
6

[Rationale]
We start with 15 trees. Later we have 21 trees. The difference must be the number of trees they planted. So, they must have planted $21 - 15 = 6$ trees.

[Example 2]
[Instruction and Question]
If there are 3 cars in the parking lot and 2 more cars arrive, how many cars are in the parking lot?

[Answer]
5

[Rationale]
There are 3 cars in the parking lot already. 2 more arrive. Now there are $3 + 2 = 5$ cars.

[Example 3]
[Instruction and Question]
Leah had 32 chocolates and her sister had 42. If they ate 35, how many pieces do they have left in total?

[Answer]
39

[Rationale]
Leah had 32 chocolates and Leah's sister had 42. That means there were originally $32 + 42 = 74$ chocolates. 35 have been eaten. So in total they still have $74 - 35 = 39$ chocolates.

[Example 4]
[Instruction and Question]
Jason had 20 lollipops. He gave Denny some lollipops. Now Jason has 12 lollipops. How many lollipops did Jason give to Denny?

[Answer]
8

[Rationale]
Jason had 20 lollipops. Since he only has 12 now, he must have given the rest to Denny. The number of lollipops he has given to Denny must have been $20 - 12 = 8$ lollipops.

[Example 5]
[Instruction and Question]
Shawn has five toys. For Christmas, he got two toys each from his mom and dad. How many toys does he have now?

[Answer]
9

[Rationale]
He has 5 toys. He got 2 from mom, so after that he has $5 + 2 = 7$ toys. Then he got 2 more from dad, so in total he has $7 + 2 = 9$ toys.

[Example 6]
[Instruction and Question]
There were nine computers in the server room. Five more computers were installed each day, from monday to thursday. How many computers are now in the server room?

[Answer]
29

[Rationale]
There are 4 days from monday to thursday. 5 computers were added each day. That means in total $4 * 5 = 20$ computers were added. There were 9 computers in the beginning, so

now there are 9 + 20 = 29 computers.

[Example 7]
[Instruction and Question]
Michael had 58 golf balls. On tuesday, he lost 23 golf balls. On wednesday, he lost 2 more. How many golf balls did he have at the end of wednesday?

[Answer]
33

[Rationale]
Michael initially had 58 balls. He lost 23 on Tuesday, so after that he has $58 - 23 = 35$ balls. On Wednesday he lost 2 more so now he has $35 - 2 = 33$ balls.

[Example 8]
[Instruction and Question]
Olivia has $23. She bought five bagels for $3 each. How much money does she have left?

[Answer]
8

[Rationale]
She bought 5 bagels for $3 each. This means she spent $5 * \$3 = \$15$ on the bagels. She had $23 in beginning, so now she has $\$23 - \$15 = \$8$.

[Example 9]
[Instruction and Question]

Prompts 4: Demonstration used for tasks in family of Arithmetic (FLAN)