# OpenReview forum: "The CoT Collection: Improving Zero-shot and Few-shot Learning of Language Models via Chain-of-Thought Fine-Tuning"
_EMNLP/2023/Conference — EMNLP 2023 Main_

### Official Review · Reviewer_qbB3 · 2023-08-01

**Soundness:** 4

**Excitement:**

3: Ambivalent: It has merits (e.g., it reports state-of-the-art results, the idea is nice), but there are key weaknesses (e.g., it describes incremental work), and it can significantly benefit from another round of revision. However, I won't object to accepting it if my co-reviewers champion it.

**Paper Topic And Main Contributions:**

The paper proposes a new CoT collection training data, which enables fine-tuning language models with billions of parameters to acquire reasoning ability. The CoT collection is first generated with human annotators and then augmented using LLMs. Experimental results on various PLMs demonstrate that the collected data is effective and can improve the models' reasoning and generalization abilities.

**Questions For The Authors:**

1. In practice, we notice that given the manually designed ICL demonstrations to LLMs and with the ground truth label concatenated, the LLMs still tend to generate rationales with incorrect intermediate inference steps, while keeping the ground truth answer as the final output. Will these bad examples affect the fine-tuning of the model, and how to ensure that the examples augmented by LLMs are completely correct?

2. See reason to reject 2


**Reasons To Accept:**

1. This paper contributes a CoT dataset that can be used to improve the reasoning and generalization abilities of the language model. The process for constructing the dataset is also proposed.
2. Extensive experiments on various PLMs demonstrate the effectiveness and correctness of the collected data, achieving notable improvement on multiple datasets.

**Reasons To Reject:**

The production of such a collection is costly, as the diversity and volume of available data is huge. Besides, the quality of rationales augmentation is hard to guarantee even with the filer method, due to the poor performance on tasks where multi-step reasoning is required. Even though the final output is correct, LLMs can still generate some intermediate inference steps that are completely wrong. [1, 2]

There is no experiment to investigate in-domain/in-task performance after CoT training (i.e. training on dataset A and evaluating on this dataset).

[1]  Scaling language models: Methods, analysis & insights from training gopher. https://arxiv.org/abs/2112.11446

[2] MathPrompter: Mathematical Reasoning using Large Language Models. https://aclanthology.org/2023.acl-industry.4.pdf

**Reproducibility:**

2: Would be hard pressed to reproduce the results. The contribution depends on data that are simply not available outside the author's institution or consortium; not enough details are provided.

**Reviewer Confidence:**

3: Pretty sure, but there's a chance I missed something. Although I have a good feel for this area in general, I did not carefully check the paper's details, e.g., the math, experimental design, or novelty.

---

> ### Author Rebuttal · Authors · 2023-08-29
>
> Thank you for your insightful feedback. **(R1) - (R2)** is our responses to 2 issues from “Reasons To Reject” and **(Q1) - (Q2)** answer the 2 questions from “Questions For The Authors”.
>
> **(R1 & Q1) Cost & Quality of Production**
> **Cost** Existing Chain of Thought (CoT) datasets, although instrumental, present limitations in terms of their scale and diversity. 9 previously available CoT datasets incorporate rationales penned by human annotators requiring a very costly and timely process [1]. Recognizing these constraints, we curate a dataset that not only increments sheer volume and diversity but also crafted with data efficiency compared to previous convention of rationale acquisition through human annotation.
>
> To address both the efficacy of the CoT Collection training and its comparative advantages relative to the 9 original CoT datasets used in Flan-T5's training, we report supplementary ablation studies. These experiments sought to evaluate zero-shot generalization capabilities under varying conditions of training instances and tasks.
>
>
> | Dataset          | Number of Tasks | Number of Instances | BBH Performance |
> |------------------|-----------------|---------------------|-----------------|
> | 9 CoT Datasets   | 9               | 180K                | 40.33           |
> | CoT Collection*  | 1060            | 10K                 | 42.04           |
> | CoT Collection** | 1060            | 100K                | 44.76           |
> | CoT Collection   | 1060            | 1.84M               | 47.60           |
>
>
> * The BBH Performance denotes the mean accuracy derived from the 23 classification datasets of BBH, as detailed in Table 2.
> * From the CoT Collection, instances were sampled in two quantities: 10K(*) and 100K(**) instances.
> * All experiments adhered to the CoT Evaluation setting.
>
> The results clearly illustrate that the enhancement in the number of tasks is the key factor for achieving robust generalization capabilities during the CoT Evaluation. Such observations are consistent with the discoveries highlighted in previous studies [2,3,4,5]. The CoT Collection's efficacy in strengthening a language model's zero-shot generalization capabilities, which was possible due to the efficient rationale collection process compared to previous human annotation methods.
>
>
> **Quality** Thank you for raising the valid concern regarding the potential presence of noisy chain of thought rationales in our augmented data set. We also acknowledge the importance of ensuring the quality and veracity of the rationales.
>
> To provide a bit more context, while the papers you referenced highlights that "the rationales might be incorrect even if the prediction is correct", our method has a slight difference. As you've pointed out in the "Questions for Authors" section, we incorporate the answer as an input during the CoT augmentation process. This procedure, though not flawless, was conceived to somewhat mitigate potential discrepancies between rationales and predictions.
>
> Automatic filtering presents significant challenges, primarily because determining the "correctness" or "factuality" within contextually-rich rationales isn't always straightforward. This issue intertwines with another pressing challenge: the adequate evaluation of such data.
>
> We genuinely value your insights and, recognizing the gravity of the concern, believe that devising methodologies to further refine and filter rationales merits significant attention and exploration in future endeavors.
>
> **(R2 & Q2) In-domain evaluation after CoT fine-tuning**
>
> Based on your suggestions, we have measured the in-domain performance using the validation split of 5 datasets (ANLI R1, ANLI R2, ANLI R3, RTE, Winogrande), which were included as training data for both Flan-T5 and CoT-T5.
>
> | **Model** | ANLI R1 | ANLI R2 | ANLI R3 | RTE   | Winogrande |
> |-----------|---------|---------|---------|-------|------------|
> | Flan-T5-11B   | 44.60    | 41.07   | 42.96   | 75.23 | 59.53      |
> | CoT-T5-11B    | 57.73   | 50.29   | 50.46   | 85.96 | 81.28      |
>
>
> Note that the evaluation was conducted using CoT Evaluation. As shown in the table above, additional CoT training significantly improves the in-domain performance. We thank you for your insight and will include the table in the camera ready version.
>
>
> *References*
>
> [1] Kim, S., Joo, S. J., Jang, Y., Chae, H., & Yeo, J. (2023, May). CoTEVer: Chain of Thought Prompting Annotation Toolkit for Explanation Verification. In Proceedings of the 17th Conference of the European Chapter of the Association for Computational Linguistics: System Demonstrations (pp. 195-208).
>
> [2] Longpre, S., Hou, L., Vu, T., Webson, A., Chung, H.W., Tay, Y., Zhou, D., Le, Q.V., Zoph, B., Wei, J., & Roberts, A. (2023). The Flan Collection: Designing Data and Methods for Effective Instruction Tuning. International Conference on Machine Learning.
>
> [3] Chung, H.W., Hou, L., Longpre, S., Zoph, B., Tay, Y., Fedus, W., Li, E., Wang, X., Dehghani, M., Brahma, S., Webson, A., Gu, S.S., Dai, Z., Suzgun, M., Chen, X., Chowdhery, A., Valter, D., Narang, S., Mishra, G., Yu, A.W., Zhao, V., Huang, Y., Dai, A.M., Yu, H., Petrov, S., Chi, E.H., Dean, J., Devlin, J., Roberts, A., Zhou, D., Le, Q.V., & Wei, J. (2022). Scaling Instruction-Finetuned Language Models. ArXiv, abs/2210.11416.
>
> [4] Ye, S., Kim, D., Jang, J., Shin, J., & Seo, M. (2022, September). Guess the instruction! flipped learning makes language models stronger zero-shot learners. In The Eleventh International Conference on Learning Representations.
>
> [5] Iyer, S., Lin, X., Pasunuru, R., Mihaylov, T., Simig, D., Yu, P., Shuster, K., Wang, T., Liu, Q., Koura, P.S., Li, X., O'Horo, B., Pereyra, G., Wang, J., Dewan, C., Celikyilmaz, A., Zettlemoyer, L., & Stoyanov, V. (2022). OPT-IML: Scaling Language Model Instruction Meta Learning through the Lens of Generalization. ArXiv, abs/2212.12017.

---

### Official Review · Reviewer_cTkT · 2023-08-02

**Typos Grammar Style And Presentation Improvements:** NA
**Soundness:** 4

**Excitement:**

4: Strong: This paper deepens the understanding of some phenomenon or lowers the barriers to an existing research direction.

**Missing References:**

NA

**Paper Topic And Main Contributions:**

The paper presents a new approach to improving the chain-of-thought (CoT) reasoning capabilities of smaller language models (LMs) by using instruction tuning with CoT rationales. To achieve this, the authors introduce a new dataset called the COT COLLECTION, which enhances the existing Flan Collection with additional rationales across various tasks. The authors demonstrate that fine-tuning smaller LMs with COT COLLECTION improves their CoT capabilities on unseen tasks. Additionally, the proposed instruction tuning approach shows promising results in few-shot learning on domain-specific tasks, even outperforming ChatGPT using demonstrations.

**Questions For The Authors:**

1) As Codex is deprecated and the CoT dataset it generated is quite large, How to find some free and readily available API options is important for future research.

2) Codex generates the CoT dataset. However, Table 3 in this paper shows the performance of other closed-source LLMs like GPT-3, Chatgpt, or GPT-4, instead of Codex's performance. Could this paper provide the difference in the performance between teacher and student models?

3) Codex is well-known for generating code or code-style CoT. However, Most instruction tasks are non-code related. Is the style of the generation by student models taught by Codex not human-like and very different from that taught by ChatGPT? Evaluating the generations from various perspectives, including diversity, fluency, etc., remains crucial and meaningful.

4) Could you clarify why the Direct evaluation results in table 2&3 are generally higher than the CoT evaluation results for the same model?

I will update the final score according to the authors responses.

**Reasons To Accept:**

1) The research topic of instruction tuning with CoT over various tasks is both new and interesting.

2) The experiment is extensive and sound.

3) The paper is well-written and well-organized.


**Reasons To Reject:**

1) This paper discusses the use of OpenAI's Codex for generating CoT, but the Codex API is now deprecated. Creating CoT data using the accessible LLM APIs would be better for ensuring reproducibility.

**Reproducibility:**

4: Could mostly reproduce the results, but there may be some variation because of sample variance or minor variations in their interpretation of the protocol or method.

**Reviewer Confidence:**

3: Pretty sure, but there's a chance I missed something. Although I have a good feel for this area in general, I did not carefully check the paper's details, e.g., the math, experimental design, or novelty.

---

> ### Author Rebuttal · Authors · 2023-08-29
>
> Thank you for your constructive feedback. Our responses for your issue from “Reasons To Reject” and 4 questions from “Questions For The Authors'' are each denoted as (R1) and (Q1)-(Q4).
>
> **(R1,Q1) Concerns regarding Reproducibility**
> We agree with your concern regarding the reproducibility since the teacher model we used for our paper (Codex) was deprecated in June of 2023. One of the main reasons for choosing Codex over ChatGPT or GPT4 was because it was available for free at the time.
>
> Nonetheless, we believe that our rationale augmentation process is not confined to Codex only, as later work such as [1] showed that using stronger teacher LMs such as GPT-4 leads to better performance on BigBench Hard. Additionally, we have released our dataset and checkpoints publicly available for future research as well.
>
> Moreover, please refer to Figure 4 in the appendix, where we used Roscoe, a suite of metrics to show that other proprietary LMs such as Bard and Claude show competitive statistics in terms of faithfulness, coherence, and informativeness of the generated CoT rationales.
>
> **(Q2) Performance of the teacher model (Codex)**
> The direct evaluation performance of Codex is 55.74, which is between ChatGPT(48.90) and GPT-4(67.40) from Table 2. We will add this result to the camera ready version of the paper.
>
> **(Q3) Concerns on Codex generation format**
> Whereas Codex was trained to specialize on code domain, it can solve different reasoning tasks by writing general chain-of-thoughts, not only code-style CoT. This is supported in the results of the [2,3] By providing ICL demonstrations written in language, we observed that Codex also generates natural CoT rationales.
>
> Additionally, we agree that thorough evaluation of CoT rationales is essential. We are keen to explore improved evaluation methods and understand their potential benefits in the future.
>
> **(Q4) Clarification on the tendency of evaluation results**
> We assume the overall dominance of Direct over CoT evaluation results comes from the fact that the volume of Flan Collection is 8 times larger than that of CoT Collection. The reduced gap between Direct and CoT of CoT-T5 compared to Flan-T5 shows that dominance of Direct evaluation can be treated by using sufficient rational data, which is what CoT Collection aims to provide. Our findings from Table 2 show that training only on CoT data could surprisingly improve the Direct evaluation performance as well, yet it is somehow ambiguous when and how positive transfer might happen between Direct evaluation and CoT evaluation.
>
> Considering the task diversity and large volume of CoT Collection compared to previous CoT datasets, we hope it could act as a stepping stone for future research exploring the impact of further expansion of rationales.
>
> *References*
>
> [1] Mukherjee, S., Mitra, A., Jawahar, G., Agarwal, S., Palangi, H., & Awadallah, A.H. (2023). Orca: Progressive Learning from Complex Explanation Traces of GPT-4. ArXiv, abs/2306.02707.
>
> [2] Wei, J., Wang, X., Schuurmans, D., Bosma, M., Xia, F., Chi, E., ... & Zhou, D. (2022). Chain-of-thought prompting elicits reasoning in large language models. Advances in Neural Information Processing Systems, 35, 24824-24837.
>
> [3] Zhang, Z., Zhang, A., Li, M., & Smola, A. (2022, September). Automatic Chain of Thought Prompting in Large Language Models. In The Eleventh International Conference on Learning Representations.

---

### Official Review · Reviewer_d7Vi · 2023-08-04

**Soundness:** 4

**Excitement:**

4: Strong: This paper deepens the understanding of some phenomenon or lowers the barriers to an existing research direction.

**Paper Topic And Main Contributions:**

Problem: Equipping smaller language models with the chain-of-thought reasoning capabilities.

Main Idea: Curate large CoT style dataset and CoT fine-tune small LLMs on that dataset

Details
1. Authors curate and release a large dataset called CoT Collection. The dataset is of the form (instruction, instance, rationale, output).
2. Authors fine-tune Flan-T5 models on the CoT Collection to obtain CoT-T5 models. They then show that CoT-T5 models are superior to Flan-T5 models in zero and few-shot setting.
3. CoT Collection is created by prompting a LLM with (instance, output) to generate a rationale. Rationales are collected for 1k+ tasks with each task assigned a task group. Demonstrations for each task group are used to prompt Codex LLM to generate the rationale for the tasks in that group.
4. Authors benchmark CoT-T5 models on BBH benchmark and compare them with models of similar sizes and commercial GPTs. Authors also do an ablation study and experiment across five different languages.

**Questions For The Authors:**

1. In multi-lingual evaluation, authors fine-tune CoT models using the data from the target language. Did authors run an experiment where CoT T5 is trained on data from all the target languages combined? Did authors observe any cross language transfer?
2. Did authors evaluate CoT-T5 models in ICL setting? How do they fare compared for commercial models?
3. Have authors checked for what datasize do the CoT T5 models start outperforming Flan-T5?

**Reasons To Accept:**

1. CoT Collection dataset is an important contribution. It will allow academics to conduct research on chain-of-thought and LMs without relying on commercial LMs.
2. The paper also presents few good insights - a) CoT fine tuning can improve model performance for direct evaluation as well b) CoT fine tuned models complement parameter efficient FT like Lora in low resource setting.

**Reasons To Reject:**

The paper points out 9 other publicly available datasets with rationale. The paper lacks discussion on the size/nature/quality of these datasets. Further, the paper does not compare effectiveness of CoT Collection dataset against other datasets. For instance, comparing performance of a CoT model trained on CoT Collection and one or more of the other datasets would have been helpful.

**Reproducibility:**

2: Would be hard pressed to reproduce the results. The contribution depends on data that are simply not available outside the author's institution or consortium; not enough details are provided.

**Reviewer Confidence:**

4: Quite sure. I tried to check the important points carefully. It's unlikely, though conceivable, that I missed something that should affect my ratings.

---

> ### Author Rebuttal · Authors · 2023-08-29
>
> Thank you for your valuable review. (R1) is our response to “Reasons To Reject” and (Q1) - (Q3) answer the 3 questions from “Questions For The Authors”.
>
>
> **(R1 & Q3) Effectiveness of CoT Collection**
>
> The 9 publicly accessible CoT datasets have been integrated into the Flan Collection and also the CoT Collection.
>
> For details on the nature of the datasets:
>
> * GSM8K and AQuA are arithmetic-focused datasets.
> * StrategyQA, ECQA, CREAK, and SenseMaking fall under the category of Multiple Choice QA datasets.
> * QASC and QED are distinguished as Extractive QA datasets.
> * e-SNLI operates as an NLI dataset.
>
> In total, these datasets consist of 180K instances. Our sampling method drew 36K instances, mirroring the dataset ratio adopted in references [1,2]. Note that all 9 datasets incorporate rationales penned by human annotators but the limited scope of these tasks result in a narrow coverage. We recognize the need of clarification upon of this information and will ensure its inclusion in the camera-ready version.
>
> Furthermore, to address both the efficacy of the CoT Collection training and its comparative advantages relative to the 9 original CoT datasets used in Flan-T5's training, we have conducted supplementary ablation studies. These experiments evaluate zero-shot generalization capabilities under varying conditions of training instances and tasks.
>
>
> | Dataset          | Number of Tasks | Number of Instances | BBH Performance |
> |------------------|-----------------|---------------------|-----------------|
> | 9 CoT Datasets   | 9               | 180K                | 40.33           |
> | CoT Collection*  | 1060            | 10K                 | 42.04           |
> | CoT Collection** | 1060            | 100K                | 44.76           |
> | CoT Collection   | 1060            | 1.84M               | 47.60           |
>
>
> * The BBH Performance denotes the mean accuracy derived from the 23 classification datasets of BBH, as detailed in Table 2.
> * From the CoT Collection, instances were sampled in two quantities: 10K(*) and 100K(**) instances.
> * All experiments adhered to the CoT Evaluation setting.
>
> The results clearly illustrate that the enhancement in the number of tasks is an important factor in achieving robust generalization capabilities for CoT Evaluation. Such observations are consistent with discoveries highlighted in previous studies [1,2,3,4]. This affirms the CoT Collection's efficacy in strengthening a language model's zero-shot generalization capabilities, even with a limited number of instances.
>
>
> **(Q1) Cross Language Transfer**
>
> Regarding the multilingual experiments:
>
> * We limited our testing to training on an individual language, without exploring cross-lingual transfer amongst varied languages.
> * The chief objective of this experimentation was to ascertain if introducing a minimal volume of CoT data could facilitate effective adaptation to the target language, specifically when addressing reasoning challenges, such as those posed by the MGSM dataset.
>
>
> Past studies have shown, in expansive models, the competency to articulate chains of thought in an array of languages is an emergent ability [5]. However, up until now, no discourse or hypothesis has suggested that training with rationales in various languages could enable cross-lingual transfers among different languages. As per the outcomes presented in our paper, we identify this as a promising avenue for future exploration.
>
>
> **(Q2) In-context Learning for CoT-T5**
>
> To briefly examine the in-context learning capabilities of CoT-T5, we selected two tasks from the Big Bench Hard dataset: Sports Understanding and Date Understanding. Few-shot CoT demonstrations from the in the original CoT study were used [6].
>
> Here are the nature of these tasks:
>
> * Date Understanding: This task necessitates reasoning that's contingent upon the provided context. It challenges the model's ability to infer and extrapolate based on given data points.
>
> * Sports Understanding: This is a more knowledge-intensive endeavor. It not only tests the model's current understanding but also its capacity to tap into foundational and background knowledge about sports.
>
>
> |        | FLAN-T5 | CoT-T5 | UL2  | LaMDA | LaMDA | LaMDA | LaMDA | LaMDA | GPT3 | GPT3 | GPT3 | GPT3 | Codex | PALM | PALM | PALM |
> |--------|---------|--------|------|-------|-------|-------|-------|-------|------|------|------|------|-------|------|------|------|
> |        | 11B     | 11B    | 20B  | 420M  | 2B    | 8B    | 68B   | 137B  | 350M | 1.3B | 6.7B | 175B | -     | 8B   | 62B  | 540B |
> | Date Understanding  | 54.8    | 64.8   | 14.0 | 1.6   | 6.8   | 5.4   | 18.6  | 26.8  | 0.9  | 1.4  | 4.9  | 52.1 | 64.8  | 13.1 | 44.7 | 65.3 |
> | Sports Understanding | 66.4    | 65.2   | 65.3 | 49.7  | 57.5  | 52.1  | 77.5  | 85.8  | 41.6 | 26.9 | 4.4  | 82.4 | 98.5  | 75.2 | 93.6 | 95.4 |
>
> Our findings indicate the following:
>
> * CoT fine-tuning noticeably augments the ICL performance when applied to Date Understanding (reasoning task).
>
> * However, this enhancement isn't ubiquitously evident. Particularly, Sports Understanding (knowledge-intensive task) don't consistently benefit from CoT fine-tuning.
>
> * Comparative Performance: Notably, CoT-T5's performance is in line with some of the more advanced, proprietary LMs available in the current landscape.
>
> Considering these outcomes, an intriguing avenue for future research lies in evaluating the potential of CoT fine-tuning to amplify the few-shot CoT prompting capabilities.
>
>
> *References*
>
> [1] Longpre, S., Hou, L., Vu, T., Webson, A., Chung, H.W., Tay, Y., Zhou, D., Le, Q.V., Zoph, B., Wei, J., & Roberts, A. (2023). The Flan Collection: Designing Data and Methods for Effective Instruction Tuning. International Conference on Machine Learning.
>
> [2] Chung, H.W., Hou, L., Longpre, S., Zoph, B., Tay, Y., Fedus, W., Li, E., Wang, X., Dehghani, M., Brahma, S., Webson, A., Gu, S.S., Dai, Z., Suzgun, M., Chen, X., Chowdhery, A., Valter, D., Narang, S., Mishra, G., Yu, A.W., Zhao, V., Huang, Y., Dai, A.M., Yu, H., Petrov, S., Chi, E.H., Dean, J., Devlin, J., Roberts, A., Zhou, D., Le, Q.V., & Wei, J. (2022). Scaling Instruction-Finetuned Language Models. ArXiv, abs/2210.11416.
>
> [3] Ye, S., Kim, D., Jang, J., Shin, J., & Seo, M. (2022, September). Guess the instruction! flipped learning makes language models stronger zero-shot learners. In The Eleventh International Conference on Learning Representations.
>
> [4] Iyer, S., Lin, X., Pasunuru, R., Mihaylov, T., Simig, D., Yu, P., Shuster, K., Wang, T., Liu, Q., Koura, P.S., Li, X., O'Horo, B., Pereyra, G., Wang, J., Dewan, C., Celikyilmaz, A., Zettlemoyer, L., & Stoyanov, V. (2022). OPT-IML: Scaling Language Model Instruction Meta Learning through the Lens of Generalization. ArXiv, abs/2212.12017.
>
> [5] Shi, F., Suzgun, M., Freitag, M., Wang, X., Srivats, S., Vosoughi, S., ... & Wei, J. (2022, September). Language models are multilingual chain-of-thought reasoners. In The Eleventh International Conference on Learning Representations.
>
> [6] Wei, J., Wang, X., Schuurmans, D., Bosma, M., Xia, F., Chi, E., ... & Zhou, D. (2022). Chain-of-thought prompting elicits reasoning in large language models. Advances in Neural Information Processing Systems, 35, 24824-24837.

---

### Meta-Review · Area_Chair_11Zd · 2023-09-17

**Recommendation:** 5

**Metareview:**

Reviewers unanimously agree the CoT Collection offers a useful resource to practitioners. With the inclusion of additional experiments and clarifications on limitations, requested by the reviewers, this makes for a relevant and robust contribution.

---

### Decision · Program_Chairs · 2023-10-07

**Decision:**

Accept-Main

**Comment:**

Reviewers unanimously agree the CoT Collection offers a useful resource to practitioners. With the inclusion of additional experiments and clarifications on limitations, requested by the reviewers, this makes for a relevant and robust contribution.